# A PROBABILISTIC FRAMEWORK FOR TASK-ALIGNED INTRA- AND INTER-AREA NEURAL MANIFOLD ESTIMATION

**Edoardo Balzani, Jean Paul Noel,**
**Pedro Herrero-Vidal, & Dora E. Angelaki**[*]
Center for Neural Science
New York University
New York, NY, 10003
`{eb162,jpn5,pmh314,da93}@nyu.edu`

**Cristina Savin** [†]
Center for Neural Science
Center for Data Science
New York University
New York, NY, 10003
`cs5360@nyu.edu`

## ABSTRACT

Latent manifolds provide a compact characterization of neural population activity and of shared co-variability across brain areas. Nonetheless, existing statistical tools for extracting neural manifolds face limitations in terms of interpretability of latents with respect to task variables, and can be hard to apply to datasets with no trial repeats. Here we propose a novel probabilistic framework that allows for interpretable partitioning of population variability within and across areas in the context of naturalistic behavior. Our approach for task aligned manifold estimation (TAME-GP) explicitly partitions variability into private and shared sources which can themselves be subdivided in task-relevant and task irrelevant components, uses a realistic Poisson noise model, and introduces temporal smoothing of latent trajectories in the form of a Gaussian Process prior. This TAME-GP graphical model allows for robust estimation of task-relevant variability in local population responses, and of shared co-variability between brain areas. We demonstrate the efficiency of our estimator on within model and biologically motivated simulated data. We also apply it to several datasets of neural population recordings during behavior. Overall, our results demonstrate the capacity of TAME-GP to capture meaningful intra- and inter-area neural variability with single trial resolution.

## 1 INTRODUCTION

Systems neuroscience is gradually shifting from relatively simple and controlled tasks, to studying naturalistic closed-loop behaviors where no two observations (i.e.,"trials") are alike (Michaiel et al., 2020; Noel et al., 2021). Concurrently, neurophysiological techniques are advancing rapidly (Stevenson & Kording, 2011; Angotzi et al., 2019; Boi et al., 2020) to allow recording from an ever-increasing number of simultaneous neurons (i.e., "neural populations") and across multiple brain areas. These trends lead to a pressing need for statistical tools that compactly characterize the statistics of neural activity within and across brain regions. Dimensionality reduction techniques are a popular tool for interrogating the structure of neural responses (Cunningham & Byron, 2014). However, as neural responses are driven by increasingly complex task features, the main axes of variability extracted using these techniques often intermix task and nuisance variables, making them hard to interpret. Alternatively, dimensionality reduction techniques that do allow for estimating task-aligned axes of variability (Brendel et al., 2011; Semedo et al., 2019; Keeley et al., 2020; Glaser et al., 2020; Hurwitz et al., 2021), do not apply to communication between brain areas, and/or necessitate trial repeat structure that does not occur in natural behavior.

Here, we introduce a probabilistic approach for learning interpretable task-relevant neural manifolds that capture both intra- and inter-area neural variability with single trial resolution. Task Aligned Manifold Estimation with Gaussian Process priors (TAME-GP) incorporates elements of demixed

---

[*]Webpage: https://angelakilabnyu.org/
[†]Webpage: http://www.cns.nyu.edu/ csavin

PCA (dPCA; Machens (2010); Kobak et al. (2016)) and probabilistic canonical correlation analysis (pCCA; Bach & Jordan (2005))[1] into a graphical model that additionally includes biologically relevant Poisson noise. The model uses a Gaussian Process (GP) prior to enforce temporal smoothness, which allows for robust reconstruction of single-trial latent dynamics (see Damianou et al. (2016) for a similar approach using Gaussian observation noise). We demonstrate the robustness and flexibility of TAME-GP in comparison to alternative approaches using synthetic data and neural recordings from rodents and primates during naturalistic tasks. This reveals TAME-GP as a valuable tool for dissecting sources of variability within and across brain areas during behavior.

**Related work.** Dimensionality reduction is usually achieved by unsupervised methods that identify axes of maximal variability in the data, such as PCA. In neuroscience, this is often accompanied by additional smoothing over time reflecting the underlying neural dynamics (e.g., Gaussian process factor analysis (GPFA) (Yu et al., 2008); see GP-LVM (Ek & Lawrence, 2009) for similar approaches outside of neuroscience). This low dimensional projection is followed by a *post hoc* interpretation of latents in the context of behavioral variables, often by visualization. Alternative approaches such as dPCA (Machens, 2010; Kobak et al., 2016) explicitly look for axes of neural variability that correlate with task variables of interest (see also Zhou & Wei (2020) for a nonlinear version). However, these require partitioning trials into relatively few categories, based on experimental conditions or behavioral choices and averaging within conditions. This makes them unusable in naturalistic tasks where a single trial treatment is needed. Similarly, SNP-GPFA (Keeley et al., 2020) can partition (multi-region) neural activity into 'shared signal' and 'private noise' components, but only using data with stimulus repeats. Under 'no-repeat' conditions, pCCA (Bach & Jordan, 2005) can find subspaces of maximal cross-correlation between linear projections of task variables and neural responses (under gaussian noise assumptions), without the need for *a priori* grouping of trials by experimental condition or choice. This approach can also be applied for determining shared axes of co-variability across areas, an analog for communication subspaces (Semedo et al., 2019). Nonetheless, its noise model assumptions are mismatched to neural data. More fundamentally, pCCA only considers pairwise relationships, preventing a joint multi-area and task variables analysis. Overall, existing approaches come with practical limitations and do not directly address the routing of task-relevant information across brain areas.

## 2 TASK-ALIGNED MANIFOLD ESTIMATION WITH GP PRIORS (TAME-GP)

In its most general form, the graphical model of TAME-GP models a set of spike-count population responses $\mathbf{x}^{(j)}$ from up to $n$ different areas,[2] together with task variable of interest $\mathbf{y}$ (Fig. 1A). The neural responses are driven by a set of $n+1$ low-dimensional latent variables $\mathbf{z}^{(j)}$. Specifically, the responses of neuron $i$ in area $j$ arise as a linear combination of private latent variability $\mathbf{z}^{(j)}$ and shared latents $\mathbf{z}^{(0)}$, which reflect task interpretable aspects of the underlying dynamics, with Poisson noise and an exponential link function:

$$p\left(\mathbf{x}_i^{(j)}|\mathbf{z}^{(0:n)}\right) = \text{Poisson}\left(\exp\left(W_i^{(0,j)}\mathbf{z}^{(0)} + W_i^{(j,j)}\mathbf{z}^{(j)} + h_i^{(j)}\right)\right), \tag{1}$$

with parameters $\mathbf{W}^{(0/j,j)}$ and $\mathbf{h}^{(j)}$.

To make latents interpretable with respect to task variables $\mathbf{y}$, we adapt a probabilistic framing of CCA (Bach & Jordan, 2005) to introduces dependencies between any of the latents $\mathbf{z}^{(k)}$), which could be private or shared across areas, and $\mathbf{y}$:

$$p\left(\mathbf{y}|\mathbf{z}^{(0)}\right) = \mathcal{N}\left(\mathbf{y}; \mathbf{C}\mathbf{z}^{(0)} + \mathbf{d}, \mathbf{\Psi}\right), \text{ with parameters } \mathbf{C}, \ \mathbf{d}, \ \mathbf{\Psi}. \tag{2}$$

---

[1]See Appendix A.1 for background on probabilistic PCA, CCA and their relation to TAME-GP.

[2]Variables $\mathbf{x}^{(j)}$, $\mathbf{y}$ are tensors with dimensions corresponding to 1) an area-specific number of neurons/ task variable dimension, 2) time within trial, and 3) trial index. We make indices explicit only where strictly needed.

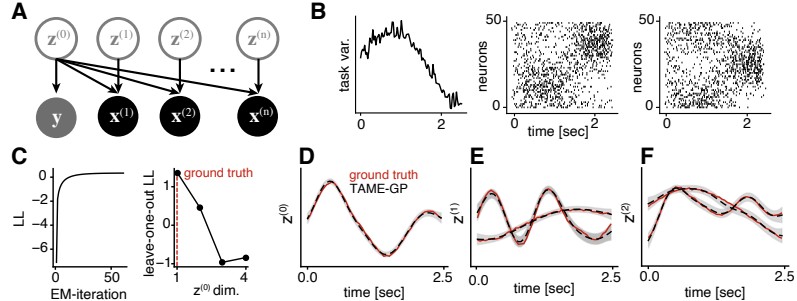

Figure 1: **A**. TAME-GP generative model. $\mathbf{z}^{(0)}$ denotes shared latent dimensions while $\mathbf{z}^{(i)}$ denote private latents of the corresponding area $i$; $\mathbf{y}$ denotes the task variables. **B**. Example draws of spiking activity and a task variable from the TAME-GP graphical model. **C**. Model log-likelihood as a function of the EM iteration (left) and cross-validated leave-one-neuron-out marginal likelihood as a function of $\mathbf{z}^{(0)}$ dimension (right). **D-F**. Latent variables estimation for within model simulated data: ground truth latent factors and model posterior mean $\pm 95\%$ CI for three latent dimensions.

Finally, we regularize all latents to be smooth over time, through the introduction of a Gaussian Process prior, as in GPFA (Yu et al., 2008),

$$z^{(j)} \sim \mathrm{GP}\left(\mathbf{0}, k_j\left(\cdot, \cdot\right)\right),, \tag{3}$$

$$k_j\left(z_{t,i}^{(j)}, z_{t',i'}^{(j)}\right) = \delta_{ii'} \exp\left(-\frac{(t-t')^2}{2\tau_i^{(j)}}\right), \tag{4}$$

with area and dimension specific hyperparameters $\tau$, $z_{t,i}^{(j)}$ is the $i$-th component of the $j$-th latent at time $t$, and $\delta_{ii'}$ is the Kronecker delta.

Putting these elements together results in a factorization of the joint distribution of the form, $p\left(\mathbf{x}^{(1:n)}, \mathbf{y}, \mathbf{z}^{(0:n)}\right) = \prod_{j=0}^{n} p\left(\mathbf{z}^{(j)}\right) p\left(\mathbf{y}|\mathbf{z}^{(0)}\right) \prod_{i,j} p\left(x_i^{(j)}|\mathbf{z}^{(0)}, \mathbf{z}^{(j)}\right)$ .This general form allows for a unified mathematical treatment of several estimation tasks of interest. We will detail key instances of this class that have practical relevance for neuroscience when presenting our numerical results below.

## 3 EM-BASED PARAMETER LEARNING

**E-step** Since a closed form solution of the posterior is not available (due to the Poisson noise), we construct a Laplace approximation of the posterior [3], $p\left(\mathbf{z}|\mathbf{x}, \mathbf{y}, \boldsymbol{\theta}\right) \approx q\left(\mathbf{z}|\mathbf{x}, \mathbf{y}, \boldsymbol{\theta}\right) = \mathcal{N}\left(\mathbf{z}; \hat{\mathbf{z}}, -\mathbf{H}^{-1}\right)$, where $\hat{\mathbf{z}}$ is the MAP of the joint log-likelihood and $\mathbf{H}$ is its corresponding Hessian. Both of these quantities are estimated numerically.

The MAP estimate is obtained by gradient descent on the joint log likelihood. The gradient of the joint log likelihood w.r.t. the latents can be written as

$$\nabla_{\mathbf{z}^{(j)}} \log p\left(\mathbf{z}, \mathbf{x}, \mathbf{y}\right) = \sum_l \left(\sum_{j \geq 0} \nabla_{\mathbf{z}^{(j)}} \log p\left(\mathbf{z}^{(j)}\right) + \sum_{t>0} \nabla_{\mathbf{z}^{(j)}} \log p\left(\mathbf{y}_t|\mathbf{z}_t^{(0)}\right)\right.$$
$$\left. + \sum_{t>0} \sum_{j>0} \nabla_{\mathbf{z}^{(j)}} \log p\left(\mathbf{x}_t^{(j)}|\mathbf{z}_t^{(0)}, \mathbf{z}_t^{(j)}\right)\right),$$

---

[3]We group latents in $\mathbf{z}$, spike counts in $\mathbf{x}$ and $\boldsymbol{\theta} = \left\{\mathbf{W}^{(0/j,j)}, \mathbf{h}^{(j)}, \mathbf{C}, \mathbf{d}, \boldsymbol{\Psi}, \tau^{(j)}\right\}$, to simplify notation.

where $l \in (1 : M)$ refers to the trial number, explicit index omitted for brevity. For a given trial, expanding one term at the time we have

$$\nabla_{\mathbf{z}^{(j)}} \log p\left(\mathbf{z}^{(j)}\right) = -\mathbf{K}^{(j)} \mathbf{z}^{(j)}$$

$$\nabla_{\mathbf{z}_t^{(0)}} \log p\left(\mathbf{y}|\mathbf{z}_t^{(0)}\right) = \mathbf{C}^\top \Psi^{-1}\left(\mathbf{y}_t - \mathbf{C}\mathbf{z}_t^{(0)} - \mathbf{d}\right)$$

$$\nabla_{\mathbf{z}_t^{(k)}} \log p\left(\mathbf{x}_t^{(j)}|\mathbf{z}_t^{(0)}, \mathbf{z}_t^{(j)}\right) = \mathbf{W}^{(k,j)\top}\left(\mathbf{x}_t - \exp\left(\mathbf{W}^{(0,j)}\mathbf{z}_t^{(0)} + \mathbf{W}^{(j,j)}\mathbf{z}_t^{(j)} + \mathbf{h}^{(j)}\right)\right),$$

where $j > 0$, $k \in \{0, j\}$ and $\mathbf{K}^{(j)}$ the GP-prior covariance matrix (Eq. 3). The corresponding second moments are

$$\nabla_{\mathbf{z}^{(j)}}^2 \log p\left(\mathbf{z}^{(j)}\right) = -\mathbf{K}^{(j)} \; j \in (0 : n)$$

$$\nabla_{\mathbf{z}_t^{(0)}}^2 \log p\left(\mathbf{y}|\mathbf{z}_t^{(0)}\right) = -\mathbf{C}^\top \Psi^{-1} \mathbf{C}$$

$$\nabla_{\mathbf{z}_t^{(h)}} \nabla_{\mathbf{z}_t^{(k)}} \log p\left(\mathbf{x}_t^{(j)}|\mathbf{z}_t^{(0)}, \mathbf{z}_t^{(j)}\right) = -\mathbf{W}^{(k,j)\top} \text{diag}\left(\exp\left(\mathbf{W}^{(0,j)}\mathbf{z}_t^{(0)} + \mathbf{W}^{(j,j)}\mathbf{z}_t^{(j)} + \mathbf{h}^{(j)}\right)\right)\mathbf{W}^{(h,j)}.$$

with $h, k \in \{0, j\}$. Inverting the $D \times D$ dimensional Hessian matrix is cubic in $D = T\sum_j d_j$, where $T$ is the trial length and $d_j$ denotes the dimensionality of latent $\mathbf{z}^{(j)}$, which restricts the number and dimensionality of latents in practice. The Hessian of the log likelihood is sparse but does not have a factorized structure. Nonetheless, we can take advantage of the block matrix inversion theorem, to speed up the computation to $\mathcal{O}(T^3 \sum_j d_j^3)$ (see Appendix A.2), with additional improvements based on sparse GP methods (Wilson & Nickisch, 2015) left for future work.

**M-step** Given the approximate posterior $q$ found in the E-step, the parameters updates can be derived analytically for a few parameters, and numerically for the rest (see Suppl. Info. A.3 for details). The other observation model parameters are computed numerically by optimizing the expected log-likelihood under the posterior. In particular, for neuron $i$ in population $j$ we have

$$\mathcal{L}\left(W_i^{(0,j)}, W_i^{(j,j)}, h_i\right) = \sum_{t,l} x_{ti}\left(h_i + \begin{bmatrix} W_i^{(0,j)} & W_i^{(j,j)} \end{bmatrix} \begin{bmatrix} \boldsymbol{\mu}_t^{(0)} \\ \boldsymbol{\mu}_t^{(j)} \end{bmatrix}\right) - \exp\left(h_i\right.$$

$$+ \begin{bmatrix} W_i^{(0,j)} & W_i^{(j,j)} \end{bmatrix} \begin{bmatrix} \boldsymbol{\mu}_t^{(0)} \\ \boldsymbol{\mu}_t^{(j)} \end{bmatrix} \frac{1}{2} \begin{bmatrix} W_i^{(0,j)} & W_i^{(j,j)} \end{bmatrix} \begin{bmatrix} \boldsymbol{\Sigma}_t^{(0,0)} & \boldsymbol{\Sigma}_t^{(0,j)} \\ \boldsymbol{\Sigma}_t^{(0,j)\top} & \boldsymbol{\Sigma}_t^{(j,j)} \end{bmatrix} \begin{bmatrix} W_i^{(0,j)\top} \\ W_i^{(j,j)\top} \end{bmatrix}\right). \quad (5)$$

For each neural population, we jointly optimized the projection weights and the intercept of all neurons with a full Newton scheme by storing the inverse Hessian in compressed sparse row (CSR) format (see Appendix A.4 for the gradient and Hessian of $\mathcal{L}$).

The GP-prior parameters were also learned from data by gradient based optimization (using the limited-memory Broyden–Fletcher–Goldfarb–Shanno scheme (Virtanen et al., 2020)). First, we set $\lambda_i^{(j)} = -\log(2\tau_i^{(j)})$, and optimize for $\lambda_i^{(j)}$ to enforce a positive time constant. We define $\boldsymbol{K}_i^{(j)} \in \mathbb{R}^{T \times T}$, such that $\left[\mathbf{K}_i^{(j)}\right]_{ts} = \exp\left(-e^{\lambda_i^{(j)}}(t-s)^2\right)$. The resulting objective function will take the form, $\mathcal{L}\left(\lambda_i^{(j)}\right) = -\text{trace}\left(\boldsymbol{K}_i^{(j)-1}\mathbb{E}_q[\boldsymbol{z}_i^{(j)}\boldsymbol{z}_i^{(j)\top}]\right) - \log|\boldsymbol{K}_i^{(j)}|$. Gradients are provided in Appendix A.5, together with the procedure for parameter initialization (Appendix A.6).

## 4 RESULTS

**Latent reconstruction for within model data.** To validate the estimation procedure, we first used a simulated dataset sampled from the TAME-GP graphical model, with predefined parameters. Specifically, we simulated two neural populations $\mathbf{x}^{(1)}$ and $\mathbf{x}^{(2)}$, each with 50 units and a one-dimensional task relevant variable $y$. We fixed the private latent factors $\mathbf{z}^{(1)}$ and $\mathbf{z}^{(2)}$ to two dimensions, and that of the shared factor $\mathbf{z}^{(0)}$ to one. The projection weights $\mathbf{W}^{(j)}$ and $\mathbf{C}$, the intercept terms $\mathbf{d}$ and $\mathbf{h}^{(j)}$, the observation variance matrix $\Phi$, and the GP time constants of the factors were randomly assigned. The parameters were chosen such that the overall mean firing rate was about 20Hz in both

areas. We simulated spike counts at 50ms resolution for 200 draws from the process (which we will refer to as 'trials' in analogy to experiments), each lasting 2.5 seconds (see example trial in Fig. 1B). Given this data, we assessed the ability of our EM-based estimator to recover its true latent structure.[4] The marginal log likelihood saturated after a relatively small number of EM iterations (Fig. 1C). As a basic test of our ability to determine the dimensionality of latents, we systematically varied the dimensionality of the shared latent, while fixing the dimensions of $\mathbf{z}^{(1)}$ and $\mathbf{z}^{(2)}$ to their ground truth value of 2. We found that the best model fit was achieved at the ground truth task dimension 1, demonstrating that we are able to infer true latent dimensionality from data (Fig.1D-F).

Finally, we assessed the quality of the recovered latents in individual test trials. Due to known degeneracies, originally documented in linear gaussian latent models (Roweis & Ghahramani, 1999), the latent factors in TAME-GP are identifiable up to an affine transformation of the latent space. To address this, we used Procustes (Schönemann, 1966) to realign the latent axes back to the original space. The resulting posterior mean estimate of the latents show an excellent agreement with the ground truth factors (cross-validated linear regression $R^2$ of 0.99 between the MAP estimate of latents and ground truth, Fig. 1 D-F), while the model predicted rates explained 98% of the ground truth firing rate variance. The ability to reconstruct ground truth structure for within model data persists when considering more than two areas with shared covariability (Suppl. Fig. S1). Overall, these numerical tests confirm that EM provides a veridical estimation of ground truth latent structure for within distribution data.

**Task-aligned latent reconstruction for simulated latent dynamical systems models.** The simple graphical model of TAME-GP captures axes of neural variability of scientific interest, but is far from an accurate generative model for neural dynamics during behavior. To assess the ability of TAME-GP to extract underlying structure from complex and out-of-distribution neural data, we used latent dynamical systems models in which we can explicitly define the flow of information from external stimuli and between areas, in several scenarios of practical interest.

The first *in silico* experiment focuses on identifying axes of task-relevant variability in neural responses. As a simple test case, we modeled a single neural population with a 6d latent structure (Fig. 2A). Two of the latent dimensions were task-relevant, driven by an observed temporally smooth external input $\mathbf{y}_t$, while the other four dimensions were intrinsic to the circuit. The key distinction between this process and the TAME-GP model assumptions is that the observed task variable acts as an input drive to the underlying latent dynamics rather than mapping to the latents directly. The latent dynamics take the form of a multivariate AR(1),

$$\begin{cases} \mathbf{z}_{\mathrm{pr},t+1} &= A_{\mathrm{pr}} \left( \mathbf{z}_{\mathrm{pr},t} - \mu_t \right) \Delta t + \sqrt{2\Delta t} \, \mathbf{dw}_t^{(0)} \\ \mathbf{z}_{\mathrm{tr},t+1} &= A_{\mathrm{tr}} \left( \mathbf{z}_{\mathrm{tr},t} - \mathbf{y}_t \right) \Delta t + \sqrt{2\Delta t} \, \mathbf{dw}_t^{(1)}, \end{cases} \tag{6}$$

where $A_{\mathrm{pr}} \in \mathbb{R}^{4\times 4}$ and $A_{\mathrm{tr}} \in \mathbb{R}^{2\times 2}$ the private and task relevant dynamics, $\mathbf{y}_t \in \mathbb{R}^2$ and $\mu_t \in \mathbb{R}^4$ inputs drawn from a factorized RBF kernel, and $\boldsymbol{w}_t^{(i)}$ is independent white noise for $i = 0, 1$. Given these latent dynamics, spikes are generated as described by the TAME-GP observation model with $\mathbf{W} \in \mathbb{R}^{100\times 6}$, and $\mathbf{d} \in \mathbb{R}^{100}$. We adjusted the parameters as to cover several average population firing rates by regulating $\mathbf{d}$, for a fixed number of trials (200) and a fixed trial duration (5 seconds). For simplicity, we circumvent the hyperparameter selection step by assuming that all estimators have access to the ground truth latent dimensionality: TAME-GP assumed 2 shared and 4 private latents. Unsupervised methods (pPCA, P-GPFA) were tasked with extracting the main two axes of neural variability in the data, while the supervised methods (pCCA) estimated 2d latents that correlate with task variable $\mathbf{y}$; the same alignment procedure was used in all cases.

Fig. 2B illustrates the latent dynamics as estimated by TAME-GP, pPCA (Tipping & Bishop, 1999), P-GPFA (Hooram, 2015), and pCCA (Bach & Jordan, 2005) . We quantify the latent space estimation accuracy by mean squared error, demonstrating that TAME-GP captured the stimulus driven dynamics better than other methods (Fig. 2C and Suppl. Fig. S2). P-GPFA showed a tendency to over-smooth, which obscured most of the underlying fine timescale latent structure. PCA failed by focusing on main axes of variability irrespective of task relevance, while CCA estimates were visually less interpretable. Only pCCA and TAME-GP found projections that selectively encoded for $\mathbf{z}_{tr}$ with TAME-GP outperforming pCCA across conditions. Finally, TAME-GP maintained its ability to

---

[4]Here and in all subsequent analyses 90% of the data is used for training the model and 10% for testing.

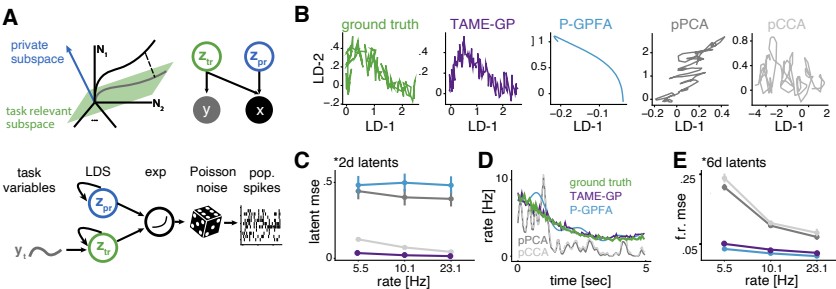

Figure 2: Methods comparison for single area task manifold alignment. **A**. TAME-GP graphical model for single area (top) and schematic for data generating process (bottom). $\mathbf{z}_{\mathrm{tr}}$ denotes task relevant shared latent dimensions while $\mathbf{z}_{\mathrm{pr}}$ denotes private task-irrelevant variability. **B**. Ground truth task relevant dynamics (green) and estimated low dimensional projection for TAME-GP (purple), P-GPFA (blue), pPCA (dark gray) and pCCA (light gray).**C** Mean squared error between the true shared dynamics and the model reconstruction, mean $\pm$ s.d. over 10-fold cross-validation. **D**. Example single trial firing rate reconstruction. **E**. Mean squared error between the true and reconstructed firing rate across conditions, mean $\pm$ s.d. over 10 folds of the data.

recover the underlying structure even when the model assumptions do not match the data exactly, in particular when the effect of the latents were modeled to be approximately additive (Suppl. Fig. S3).

We also compared these methods in terms of their ability to predict the ground truth firing rate generating the observed spiking responses (total dimensions matching the ground truth of 6). Both TAME-GP and P-GPFA showed a stable and accurate firing rate reconstruction error across conditions (Fig. 2D,E), while the factorized linear gaussian methods (pPCA, pCCA) performed poorly. This may be due to the larger model mismatch, while additionally suffering from the lack of temporal smoothing, especially for low firing rates. Overall, TAME-GP was the only procedure that both captured the overall data statistics well and extracted accurate task-interpretable latents.

**Assessing inter-area communication in simulated latent dynamical systems.** In the second set of numerical experiments, we focused on estimating low-dimensional communication sub-spaces across neural populations (Fig. 3A). The ground truth data was again constructed using latent dynamical systems models, which now included two populations (Fig. 3B), where a low dimensional projection of the dynamics in one area, the sender, drive the dynamics of the other area, the receiver:

$$\begin{cases} \mathbf{z}_{\mathrm{S},t+1} & = A_S \left( \mathbf{z}_{\mathrm{S},t} - \mathbf{y}_t \right) \Delta t + \sqrt{2\Delta t}\mathbf{w}_t^{(0)} \\ \mathbf{z}_{\mathrm{sh}} & = P \cdot \mathbf{z}_{\mathrm{S}} \\ \mathbf{z}_{\mathrm{R},t+1} & = A_R \left( \mathbf{z}_{\mathrm{R},t} - \lambda_t - \mathbf{z}_{\mathrm{sh},t} \right) \Delta t + \sqrt{2\Delta t}\mathbf{w}_t^{(1)}, \end{cases} \quad (7)$$

where $A_S \in \mathbb{R}^{4\times4}$ and $A_R \in \mathbb{R}^{4\times4}$ are the sender and receiver dynamics, $\mathbf{y}_t$ and $\lambda_t$ are temporally smooth inputs drawn from independent GPs with factorized RBF kernels, $P \in \mathbb{R}^{2\times4}$ defines the shared submanifold projection, and $w_t^{(i)}$ is independent white noise. These latents map into spikes as above. We simulated three average firing rate conditions and varied the ground truth number of shared dimensions, from one to three. We compared our method with the two most commonly used approaches to communication manifold: pCCA and Semedo's reduced-rank regression procedure for communication manifold estimation (Semedo et al., 2019) (Fig. 3C), as well as SNP-GPFA (Keeley et al., 2020) (both with and without trial repeats, see Appendix A.7 and Suppl. Fig. S4).

TAME-GP (without task alignment) outperformed alternative approaches in terms of the reconstruction error of both ground truth firing rates (Fig. 3D, F) and shared latent dynamics (Fig. 3E). Furthermore, when testing the ability of different approaches to infer the dimensionality of the shared manifold through model comparison, the leave-one-out likelihood saturated at the ground truth dimension for all simulations (Fig. 3I), and peaked at the correct dimension 75% of the times (Fig. 3G, H). In contrast, the Semedo estimator tends to systematically overestimate the dimensionality of the shared manifold in this dataset.

Finally, we tested the general case in which we search for a communication subspace that aligns to task variable $\mathbf{y}$. To do so, we fit TAME-GP to the same dataset but assuming that $\mathbf{y}_t$ is observed.

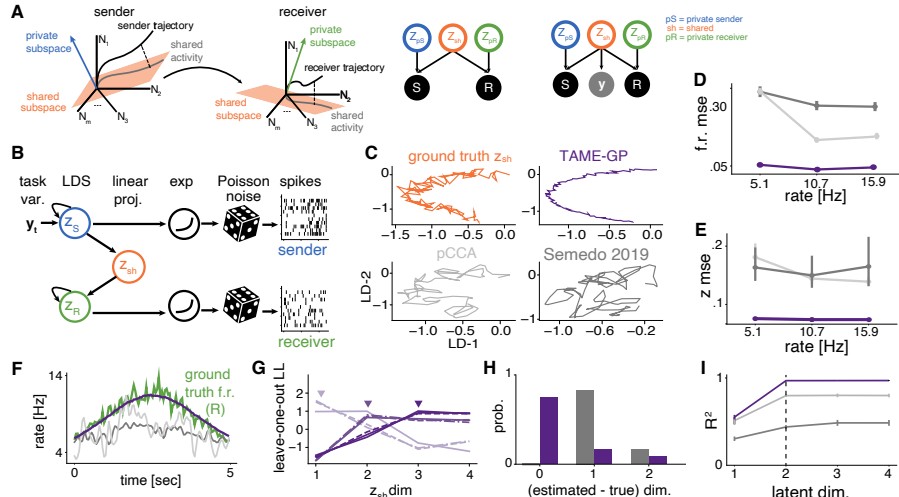

Figure 3: **A**. Schematic of communication subspace (left) and associated TAME-GP graphical model versions (right). **B**. Ground truth spike count generation process. **C**. Example shared latent reconstruction for TAME-GP (purple), PCCA (light grey) and, reduced rank regression (dark grey); ground truth in orange. **D**. Statistics for firing rate prediction quality. **E**. Statistics of shared dynamics reconstruction. **F**. Example reconstructions of the receiver firing rates compared to the ground truth (green). **G**. TAME-GP leave-one-neuron-out log-likelihood for different ground truth shared manifold dimensionality (d=1,2,3) and increasing population rate from 5.1, 10.7, 15.9 Hz (respectively, dashed, dashed-dotted and continuous lines). Lines styles show different average firing rate conditions. **H**. Difference between estimated and true $z_{sh}$ dimensionality for TAME-GP (purple) and reduced rank regression (grey). **I**. Model fit quality as a function of latent dimensionality for all estimators. Ground truth dimension d=2 (dashed line). Error bars show mean $\pm$ s.d. over 10-folds of cross-validation.

We found again that TAME-GP has the best reconstruction accuracy, which saturates at the ground truth dimensionality (d=2). These observations are consistent across firing rate levels (see Suppl. Fig. S5). When fitting SNP-GPFA to simulated data in the case of precise stimulus repetitions and comparing it to TAME-GP, we find that both models are able to capture the latent space factorization. However, only TAME-GP works well in the case when latent dynamics vary across episodes, as would be the case during natural behavior (i.e. without stimulus repeats, see Suppl. Fig. S4, Table S1 and Appendix A.7 for details). Overall, these results suggest that TAME-GP can robustly recover meaningful sources of co-variability across areas in a range of experimentally relevant setups.

**Mouse neural recordings during open-field exploration.** As a first validation of the method, we estimated the manifold formed by simultaneously recorded head direction cells (n = 33) in the anterodorsal thalamic nuclei (ADN) (Taube, 1995) of a mouse exploring a circular open field. [5]

These neurons are known to form a circular manifold representing heading direction (Chaudhuri et al., 2019). Thus, they provide the opportunity to examine the ability of TAME-GP to recover the underlying structure of data for which we know the biological ground truth. Recorded responses were segmented in 10sec time series, discretized in 20ms bins, and fit with a either a head-direction aligned 2d latent manifold (Fig.4A); private noise dimension d=5), or with two unsupervised methods pPCA and PGPFA, each with latent dimensionality d=2. All methods recovered the underlying circular structure of the heading representation to some degree (Fig.4B). We decoded head direction from the extracted 2d latents[6] and confirmed that TAME-GP preserved more information than pPCA, and comparable to P-GPFA (Fig.4C), with an overall superior data fit quality relative to pPCA (Fig.4D), as assessed by the $R^2$ between model leave-one-neuron-out firing rate predictions and the raw spike

---

[5]Surgeries and procedures were approved by the Institutional Animal Care and Use Committee at Baylor College of Medicine and New York University and were in accordance with National Institute of Health guidelines, protocol number 18-1502.

[6]Decoding was performed using Lasso regression, with hyperparameter selection by 5-fold cross-validation.

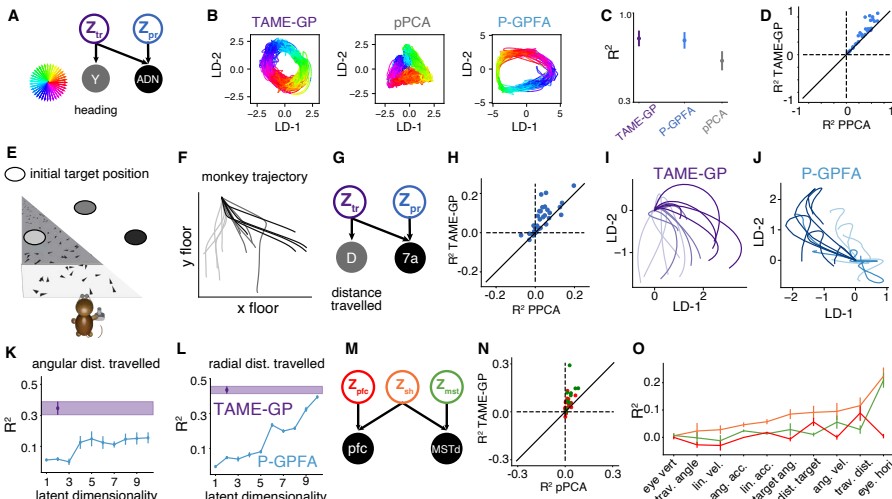

Figure 4: Fitting TAME-GP to neural data. **A**. Graphical model for heading aligned mouse population responses in area ADN. $z_{tr}$ denotes heading related shared latent dimensions while $z_{pr}$ denotes private task-irrelevant variability. **B**. Latent population dynamics, colored by time-varying heading, for various manifold estimators. **C**. Head direction decoding from 2d latents extracted with each method (by Lasso regression). Mean $\pm$ standard deviation over 5folds. **D**. Scatter plot of leave-one-neuron-out spike count variance explained for dimension matched TAME-GP and pPCA. Dots represent individual neurons. **E**. Schematic of the firefly task. Initial target location is randomized and remains visible for 300ms. The monkey has to use the joystick to navigate to the internally maintained target position. **F**. Top view of example monkey trajectories; increasing contrast marks initial location of the target (right, center, left). **G**. Within-area TAME-GP estimation aligned a latent task variable: the distance travelled. **H**. Scatter plot of leave-one-neuron-out spike count variance explained for dimension-matched TAME-GP and pPCA. Dots represent individual neurons. **I**. Single trial TAME-GP estimates of the task relevant dynamics, compared to **J**. those of P-GPFA. Trajectories are color-graded according to the initial angular target location (as in B). Lasso regression decoding of **K**. and **L**. linear distance travelled. TAME-GP decoding $R^2$ (purple) is based on a 2d task relevant latent. P-GPFA $R^2$ (blue) estimates were obtained for a range of latent dimensions (1-10). **M**. Communication subspace estimation between MSTd and dlPFC. **N**. As H, for shared latent space. **O**. Lasso regression decoding of task relevant variables (sorted by their shared subspace information content) from the shared (orange) and private latents (green, red) estimated by TAME-GP. Mean $R^2 \pm$ s.e.m. estimated across 10 folds of the data.

counts (Yu et al., 2008). Overall, these results confirm that the TAME-GP estimator can extract sensible coding structure from real data that does not exactly match the assumptions of the model.

**Multi-area neural recordings in monkeys during VR spatial navigation**   Finally, we tested the ability of TAME-GP to find task aligned neural manifolds in a challenging dataset characterized by a high-dimensional task space and lack of trial repeats. Specifically, monkeys navigate in virtual reality by using a joystick controlling their linear and angular velocity to "catch fireflies" (Fig.4E, F) (Lakshminarasimhan et al., 2018). Spiking activity was measured (binned in 6ms windows, sessions lasting over 90min) and neurons in the two recorded brain areas (MSTd and dlPFC) showed mixed selectivity, encoding a multitude of task relevant variables (Noel et al., 2021). As a result, responses are high dimensional and unsupervised dimensionality reduction methods capture an hard to interpret mixture of task relevant signals in their first few latent dimensions.

We used TAME-GP to extract latent projections that align with the ongoing distance from the origin, decomposed in an angular and a radial component (Fig. 4G). We set the task relevant latent $z^{(0)}$ dimensions to two, matching the number of task variables. We verified the accuracy of the model by computing leave-one-neuron-out firing rate predictions and calculating the $R^2$ between model predictions and raw spike counts. The TAME-GP estimator systematically outperformed pPCA with matched number of latents by this metric (Fig. 4H). We also compared the latent factors found

by TAME-GP to those obtained by P-GPFA (Fig. 4I, J). For both variables, we found that the task variables were better accounted for by a two-dimensional TAME-GP estimated latent than by up to 10 dimensional latent spaces extracted with P-GPFA (Fig. 4K, L). A similar compression of the manifold was achieved in a separate dataset of monkey (pre-)motor responses during sequential reaches (see A.9 and Suppl. Fig. S7). This confirms that TAME-GP provides a compact low dimensional account of neural variability with respect of task variables of interest.

Lastly, we probed the model's ability to learn a communication subspace (Fig. 4M) between MSTd and dlPFC, brain areas that are known to interact during this task (Noel et al., 2021). In this instance, we selected the number of shared and private latent dimensions by maximizing the leave-one-neuron-out spike counts variance explained over a grid of candidate values (see Suppl. Fig. S6 and A.8). As before, we find that the TAME-GP reconstruction accuracy surpasses that of dimensionality-matched pPCA, for both MSTd and dlPFC (Fig. 4N). Since the shared manifold estimation was agnostic to task variables in this case, we used decoding from latent spaces to ask if the shared variability between these areas carried information about task variables known to drive single neuron responses in these areas. We found that the monkey's horizontal eye position, as well as latent task variables such as the travelled distance or the distance still remaining to target were mostly accounted for in shared, as opposed to private, axes of variability (Fig. 4O). This recapitulates prior observations made at the single-cell level (Noel et al., 2021). Overall, the results demonstrate that TAME-GP can extract interpretable low-dimensional latents and shared neural subspaces from complex and high-dimensional datasets.

## 5 DISCUSSION

Technological advances in systems neuroscience place an ever-increasing premium on the ability to concisely describe high-dimensional task-relevant neural responses. While sophisticated methods based on recurrent neural networks are increasingly used for fitting neural responses Pandarinath et al. (2018), the extracted dynamics are also not necessarily easy to interpret. Here we introduce TAME-GP, a flexible statistical framework for partitioning neural variability in terms of private or shared (i.e., inter-area) sources, aligned to task variables of interest, and with single trial resolution. We show that our method provides compact latent manifold descriptions that better capture neural variability than any of the standard approaches we compared it against.

An important nuance that distinguishes various neural dimensionality reduction methods is whether the covariability being modeled is that of trial-averaged responses (i.e. stimulus correlations), residual fluctuations around mean responses (i.e. noise correlations) or a combination of the two (total correlations). Since isolating either the signal or the noise correlations alone would require across trial averages, our approach models total correlations, time resolved within individual trials. This differentiates our shared variability estimates from the traditional definition of a communication subspace (Semedo et al., 2019), which uses noise correlations alone, while keeping some of its spirit. It also makes it applicable to datasets without trial repeats.

The model adapts the approach of pCCA as a way of ensuring that the extracted latents reflect axes of neural variability that carry specific task relevant information. This choice has appealing mathematical properties in terms of unifying the problems of finding interpretable axes and communication subspaces, but is not the most natural one in terms of the true generative process of the data. While behavioral outputs can be thought of as outcomes of neural activity —as described by the TAME-GP graphical model, sensory variables act as drivers for the neural responses and should affect the latent dynamics, not the other way around. Hence a natural next step will be to incorporate in the framework explicit stimulus responses, perhaps by taking advantage of recent advances in estimating complex tuning functions during naturalistic behavior (Balzani et al., 2020).

It would be interesting to explore the use temporal priors with more interesting structure, for instance spectral mixture kernels (Wilson & Adams, 2013), introducing prior dependencies across latent dimensions (de Wolff et al., 2021), or using non-reversible GP priors that better capture the causal structure of neural dynamics (Rutten et al., 2020). More generally, the probabilistic formulation allows the ideas formalized by TAME-GP to be combined with other probabilistic approaches for describing stimulus tuning and explicit latent neural dynamics (Duncker et al., 2019; Glaser et al., 2020; Duncker & Sahani, 2021). Hence, this work adds yet another building block in our statistical arsenal for tackling questions about neural population activity as substrate for brain computation.

**Broader impact** We do not foresee any negative consequences to society from our work. Task aligned manifold extraction may prove useful in clinical applications, specifically for increasing robustness of BMI decoders by exploiting the intrinsic structure of the neural responses. Code implementing the TAME-GP estimator and associated demos is available at *https://github.com/BalzaniEdoardo/TAME-GP*

**Acknowledgements.** This work was supported by the National Institute of Health under the U19 research program (grant agreement number NIH U19NS118246).

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

## A    APPENDIX

### A.1    BACKGROUND ON PPCA, PCCA, AND THEIR RELATION TO TAME-GP

#### A.1.1    CANONICAL CORRELATION ANALYSIS

Given a random vector $\boldsymbol{x}$, PCA aims to find a linear transformation such that the components of the transformed vector are uncorrelated. In other words, it tries to find a linear transformation that diagonalizes the co-variance matrix of the random vectors. Similarly, CCA starts from two random vectors $\boldsymbol{x}_1$ and $\boldsymbol{x}_2$ of dimensions $m_1$ and $m_2$, and tries to find two linear transformations $U \in \mathbb{R}^{m_1 \times m_1}$ and $V \in \mathbb{R}^{m_2 \times m_2}$ such that each component of $U \cdot \boldsymbol{x}_1$ is correlated with a single component of $V \cdot \boldsymbol{x}_2$. In terms of correlation matrix, this corresponds to,

$$\text{corr}\left(U \cdot \boldsymbol{x}_1, V \cdot \boldsymbol{x}_2\right)_{ij} = \begin{cases} \rho_i & \text{if } i = j \\ 0 & \text{otherwise} \end{cases} \tag{8}$$

where $\rho_i$ are called canonical correlations.

Letting the joint empirical co-variance be $\hat{\Sigma} = \begin{bmatrix} \hat{\Sigma}_{11} & \hat{\Sigma}_{12} \\ \hat{\Sigma}_{21} & \hat{\Sigma}_{22} \end{bmatrix}$,

it turns out that CCA projections are the singular vectors of the correlation matrix re-scaled by the inverse square-root of the individual co-variances. Namely, if $\tilde{u}_i, \tilde{v}_i$ are the i-th singular vectors of the correlation matrix $\text{corr}\left(\boldsymbol{x}_1, \boldsymbol{x}_2\right) = \hat{\Sigma}_{11}^{-1/2} \hat{\Sigma}_{12} \hat{\Sigma}_{22}^{-1/2}$, then the canonical vectors are $(U_i, V_i) = (\hat{\Sigma}_{11}^{-1/2} \tilde{u}_i, \hat{\Sigma}_{22}^{-1/2} \tilde{v}_i)$. The two projection matrices $U$ and $V$ are obtained by stacking the canonical vectors; it is immediate to verify that $I_{m_1} = U^\top \hat{\Sigma}_{11} U$, $I_{m_2} = V^\top \hat{\Sigma}_{22} V$ and $P = U^\top \hat{\Sigma}_{12} V$, where $P$ is an $m_1 \times m_2$ diagonal matrix with diagonal entries the canonical correlations.

Again, making the parallel with PCA, we know that the first PCA vector is the eigenvector of the empirical co-variance corresponding to the largest eigenvalue and satisfies $\boldsymbol{w}_1 = \underset{\|\boldsymbol{w}\|=1}{\operatorname{argmax}} \boldsymbol{w}^\top \text{cov}(\boldsymbol{x})\boldsymbol{w}$. Similarly, it can be shown that the canonical vector corresponding to the largest singular value of the correlation matrix satisfies,

$$(U_1, V_1) = \underset{\|\boldsymbol{u}\|=1, \|\boldsymbol{v}\|=1}{\operatorname{argmax}} \text{corr}(\boldsymbol{u}^\top \cdot \boldsymbol{x}_1, \boldsymbol{v}^\top \cdot \boldsymbol{x}_2). \tag{9}$$

Finally the n-th canonical vector satisfies,

$$(U_n, V_n) = \underset{\boldsymbol{u} \in \mathcal{U}^\perp, \boldsymbol{v} \in \mathcal{V}^\perp}{\operatorname{argmax}} \text{corr}(\boldsymbol{u}^\top \cdot \boldsymbol{x}_1, \boldsymbol{v}^\top \cdot \boldsymbol{x}_2). \tag{10}$$

with $\mathcal{U}^\perp = \{\boldsymbol{u} : \|\boldsymbol{u}\| = 1, \boldsymbol{u} \in \langle U_1, \cdots U_{n-1} \rangle^\perp\}$ and $\mathcal{V}^\perp = \{\boldsymbol{v} : \|\boldsymbol{v}\| = 1, \boldsymbol{v} \in \langle V_1, \cdots V_{n-1} \rangle^\perp\}$.

A.1.2    THE PROBABILISTIC INTERPRETATION OF PCA AND CCA

As first shown by Tipping and Bishop Tipping & Bishop (1999), PCA can be expressed in terms of the maximum likelihood solution of the following probabilistic latent variable model,

$$p(\boldsymbol{z}) \sim \mathcal{N}(0, I) \tag{11}$$

$$p(\boldsymbol{x}|\boldsymbol{z}) \sim \mathcal{N}(W\boldsymbol{z} + \mu, I, \sigma^2 I), \tag{12}$$

where $I$ is the $D \times D$ identity matrix, $W$ is a $N \times D$ projection matrix, $\mu \in \mathbb{R}^N$ is an intercept term and $\sigma^2$ a positive constant.

Similarly, Bach and Jordan Bach & Jordan (2005) showed that the canonical directions emerge from the maximum likelihood estimates of a simple probabilistic model,

$$\boldsymbol{z} \sim \mathcal{N}(0, I_D) \qquad\qquad D = \min(m_1, m_2) \tag{13}$$

$$\boldsymbol{x_1}|\boldsymbol{z} \sim \mathcal{N}(W_1 \boldsymbol{z} + \mu_1, \Psi_1) \qquad \Psi_1 \succeq 0 \tag{14}$$

$$\boldsymbol{x_2}|\boldsymbol{z} \sim \mathcal{N}(W_2 \boldsymbol{z} + \mu_2, \Psi_2) \qquad \Psi_2 \succeq 0. \tag{15}$$

where we use a notation similar to that of equations (11, 12) for the projection weights, the intercept and the identity matrix, while $\Psi_1$ and $\Psi_2$ are generic positive semi-definite $N \times N$ matrices.

We will refer to these models as the probabilistic PCA and probabilistic CCA, or pPCA and pCCA.

Too better highlight the link between CCA and pCCA we report the ML estimates of the pCCA model parameters,

$$\hat{W}_1 = \hat{\Sigma}_{11} U M_1 \tag{16}$$

$$\hat{W}_2 = \hat{\Sigma}_{22} V M_1 \tag{17}$$

$$\hat{\Psi}_1 = \hat{\Sigma}_{11} - \hat{W}_1 \hat{W}_1^\top \tag{18}$$

$$\hat{\Psi}_2 = \hat{\Sigma}_{22} - \hat{W}_2 \hat{W}_2^\top \tag{19}$$

$$\hat{\mu}_1 = \frac{1}{N} \sum_j x_{1j} \tag{20}$$

$$\hat{\mu}_2 = \frac{1}{N} \sum_j x_{2j}, \tag{21}$$

where $M_i$ are arbitrary $D \times D$ matrices such that $M_1 M_2^\top = P$, the diagonal matrix of the canonical correlations, $U$ and $V$ are the canonical directions.

The posterior means and co-variances are given by,

$$\mathbb{E}[\boldsymbol{z}|\boldsymbol{x}_1] = M_1^\top U^\top (\boldsymbol{x}_1 - \hat{\mu}_1) \tag{22}$$

$$\mathbb{E}[\boldsymbol{z}|\boldsymbol{x}_2] = M_2^\top V^\top (\boldsymbol{x}_2 - \hat{\mu}_2) \tag{23}$$

$$\text{cov}\,(\boldsymbol{z}|\boldsymbol{x}_1) = I - M_1 M_1^\top \tag{24}$$

$$\text{cov}\,(\boldsymbol{z}|\boldsymbol{x}_2) = I - M_2 M_2^\top \tag{25}$$

$$\mathbb{E}[\boldsymbol{z}|\boldsymbol{x}_1, \boldsymbol{x}_2] = \begin{bmatrix} M_1 \\ M_2 \end{bmatrix} \begin{bmatrix} (I - P^2)^{-1} & (I - P^2)^{-1}P \\ (I - P^2)^{-1}P & (I - P^2)^{-1} \end{bmatrix} \begin{bmatrix} U^\top (\boldsymbol{x}_1 - \hat{\mu}_1) \\ V^\top (\boldsymbol{x}_2 - \hat{\mu}_2) \end{bmatrix} \tag{26}$$

$$\text{cov}\,(\boldsymbol{z}|\boldsymbol{x}_1, \boldsymbol{x}_2) = I - \begin{bmatrix} M_1 \\ M_2 \end{bmatrix} \begin{bmatrix} (I - P^2)^{-1} & (I - P^2)^{-1}P \\ (I - P^2)^{-1}P & (I - P^2)^{-1} \end{bmatrix} \begin{bmatrix} M_1 \\ M_2 \end{bmatrix}^\top. \tag{27}$$

It is important to notice that, independently of the $M_1$ and $M_2$ matrices, the observation gets projected into the $D$-dimensional subspace of the canonical directions. See Bishop & Nasrabadi (2006) for a similar argument bridging PCA and pPCA.

### A.1.3 TAME-GP COMBINES AND EXTENDS THE pPCA AND pCCA GENERATIVE MODELS

The probabilistic interpretation of PCA and CCA, Eqs. (11-15) - allows (1) extending the model to non-Gaussian observation noise, (2) replacing the normal prior over the latent with a smoothing GP-prior, and (3) combining the two graphical models in a more general framework.

In particular, TAME-GP assumes a shared latent factor $\boldsymbol{z}^{(0)}$ with a GP prior that captures fine time scale correlations between some continuous task variables of interest (modelled as conditionally Gaussian) and the spike counts from multiple brain regions (modelled as conditionally Poisson). This approach extends the ideas of pCCA to the analysis of spike trains driven by smooth temporal dynamics. Further, we extended our graphical model by including additional area-specific latent factors $\boldsymbol{z}^{(j)}$ (GP-distributed). The projection associated with those factors aim specifically to capture the residual inter-area co-fluctuations, in close resemblance to the role of the pPCA projection weights.

The general formulation of the TAME-GP generative model is given by Eqs.1-3 in the main text.

### A.2 INVERTING THE HESSIAN OF THE JOINT LOG-LIKELIHOOD

The dimensionality of the individual latents and trial duration pose computational challenges for TAME-GP approximate inference. For each trial, evaluating the posterior covariance requires inverting the Hessian of the joint log-likelihood, of dimensionality $D \times D$, where $D = T \sum_j d_j$, $d_j$ is the dimension of $\boldsymbol{z}^{(j)}$ and $T$ is the number of time points of the trial (for simplicity, we assume all trials are the same length here, but the implementation allows for variability in trial duration). Hence, a naive implementation of the posterior estimation would require $O\left(D^3\right)$ operations (the cost of inverting a $D$-dimensional matrix). Nonetheless, the specific conditional independence assumptions of our model allow us to speed up this computation by using the block matrix inversion theorem. In particular, if we define

$$\nabla_{\boldsymbol{z}^{(h)}} \nabla_{\boldsymbol{z}^{(k)}} \log p(\boldsymbol{z}, \boldsymbol{x}, \boldsymbol{y}) \equiv H_{hk},$$

$\boldsymbol{H}$ has the following structure,

$$H = \begin{bmatrix} H_{00} & H_{01} & H_{02} & \cdots & H_{0n} \\ H_{01}^\top & H_{11} & \boldsymbol{0} & \cdots & \boldsymbol{0} \\ H_{02}^\top & \boldsymbol{0} & H_{22} & \cdots & \boldsymbol{0} \\ & & & \ddots & \\ H_{0n}^\top & \boldsymbol{0} & \boldsymbol{0} & \cdots & H_{nn} \end{bmatrix},$$

therefore, it can be inverted according to,

$$\begin{bmatrix} A & C^\top \\ C & B \end{bmatrix}^{-1} = \begin{bmatrix} (A - C^\top B^{-1} C)^{-1} & -(A - C^\top B^{-1} C)^{-1} C^\top B^{-1} \\ -CB^{-1}(A - C^\top B^{-1} C)^{-1} & B^{-1} + B^{-1} C (A - C^\top B^{-1} C) C^\top B^{-1} \end{bmatrix},$$

by setting $A = H_{00}$ and $B = \begin{bmatrix} H_{11} & \boldsymbol{0} & \cdots & \boldsymbol{0} \\ \boldsymbol{0} & H_{22} & \cdots & \boldsymbol{0} \\ & & \ddots & \\ \boldsymbol{0} & \boldsymbol{0} & \cdots & H_{nn} \end{bmatrix}$, and $C = \begin{bmatrix} H_{01}^\top \\ \vdots \\ H_{0n}^\top \end{bmatrix}$; computing $B^{-1}$ requires only inverting the block-diagonal elements, while $(A - C^\top B^{-1} C)$ has the same size as $H_{00}$, achieving an inversion of $\boldsymbol{H}$ in $O(T^3 \sum_j d_j^3)$ operations.

### A.3 PARAMETER UPDATE DETAILS

Introducing the notation $\boldsymbol{\mu}_t^{(k)} = \mathbb{E}_q[\boldsymbol{z}_t^k]$ and $\boldsymbol{\Sigma}_t^{(k,h)} = \mathbb{E}_q[\boldsymbol{z}_t^{(k)}\boldsymbol{z}_t^{(h)\top}] - \boldsymbol{\mu}_t^{(k)}\boldsymbol{\mu}_t^{(h)\top}$, we have

$$\bar{\mathbf{C}} = \left[\sum_{l,t}\mathbf{y}_t\boldsymbol{\mu}_t^{(0)\top} - \frac{1}{TM}\sum_{l,t}\mathbf{y}_t\sum_{l,t}\boldsymbol{\mu}_t^{(0)\top}\right]\left[\sum_{l,t}\boldsymbol{\Sigma}_t^{(0,0)} + \sum_{l,t}\boldsymbol{\mu}_t^{(0)}\boldsymbol{\mu}_t^{(0)\top} - \frac{1}{TM}\sum_{l,t}\boldsymbol{\mu}_t^{(0)}\sum_{l,t}\boldsymbol{\mu}_t^{(0)\top}\right]^{-1}$$

$$\bar{\mathbf{d}} = \frac{1}{TM}\left(\sum_{l,t}\mathbf{y}_t - \bar{\mathbf{C}}\sum_{l,t}\boldsymbol{\mu}_t^{(0)}\right)$$

$$\bar{\Psi} = \frac{1}{TM}\left[\sum_{l,t}\mathbf{y}_t\mathbf{y}_t^\top - \left(\sum_{l,t}\mathbf{y}_t\boldsymbol{\mu}_t^{(0)\top}\bar{\mathbf{C}}^\top + \bar{\mathbf{C}}\sum_{l,t}\boldsymbol{\mu}_t^{(0)}\mathbf{y}_t^\top\right) - \left(\sum_{l,t}\mathbf{y}_t\bar{\mathbf{d}}^\top + \bar{\mathbf{d}}\sum_{l,t}\mathbf{y}_t^\top\right)\right.$$

$$\left. + \bar{\mathbf{C}}\left(\sum_{l,t}(\boldsymbol{\Sigma}_t^{(0,0)} + \boldsymbol{\mu}_t\boldsymbol{\mu}_t^{(0)})\right)\bar{\mathbf{C}}^\top + \left(\bar{\mathbf{C}}\sum_{l,t}\boldsymbol{\mu}_t^{(0)}\bar{\mathbf{d}}^\top + \bar{\mathbf{d}}\sum_{l,t}\boldsymbol{\mu}_t^{(0)\top}\bar{\mathbf{C}}^\top\right) + TM\bar{\mathbf{d}}\bar{\mathbf{d}}^\top\right]$$

where $l = 1 : M$ and $t = 1 : T$ are trial and time within trial indices.

### A.4 LEARNING THE POISSON OBSERVATION PARAMETERS

In order to learn the Poisson observation parameters we numerically maximize $\mathbb{E}_q\left[\log(p(\mathbf{x}, \mathbf{y}, \mathbf{z}|\boldsymbol{\theta})\right]$ as a function of $W^{(0,j)}$, $W^{(j,j)}$ and $\mathbf{h}^{(j)}$ [7]. Our implementation follows a Newton scheme which requires both the gradient and the Hessian of the optimization objective.

In order to simplify notation, we fix a unit $i$ from population $j$ and we set

$$\boldsymbol{\mu}_t = \begin{bmatrix}\boldsymbol{\mu}_t^{(0)}\\\boldsymbol{\mu}_t^{(j)}\end{bmatrix}$$

$$\Sigma_t = \begin{bmatrix}\Sigma_t^{(0,0)} & \Sigma_t^{(0,j)}\\\Sigma_t^{(0,j)\top} & \Sigma_t^{(j,j)}\end{bmatrix}$$

$$W = \begin{bmatrix}W_i^{(0,j)\ \top}\\W_i^{(j,j)\ \top}\end{bmatrix}$$

$$x_t = x_{it}^{(j)}$$

$$h = h_i^{(j)},$$

where $W \in \mathbb{R}^{d_0 + d_j}$, and $h \in \mathbb{R}$. The corresponding gradient and derivative will be,

$$\frac{\partial\mathbb{E}_q\left[\log(p(\mathbf{x}, \mathbf{y}, \mathbf{z}|\boldsymbol{\theta})\right]}{\partial W} = \sum_{l,t}x_t\boldsymbol{\mu}_t - e^{h + W^\top\boldsymbol{\mu}_t + \frac{1}{2}W^\top\Sigma_t W}\left(\boldsymbol{\mu}_t + \Sigma_t W\right) \tag{28}$$

$$\frac{\partial\mathbb{E}_q\left[\log(p(\mathbf{x}, \mathbf{y}, \mathbf{z}|\boldsymbol{\theta})\right]}{\partial h} = \sum_{l,t}x_t - e^{h + W^\top\boldsymbol{\mu}_t + \frac{1}{2}W^\top\Sigma_t W} \tag{29}$$

$$\frac{\partial^2\mathbb{E}_q\left[\log(p(\mathbf{x}, \mathbf{y}, \mathbf{z}|\boldsymbol{\theta})\right]}{\partial W^2} = -e^{h + W^\top\boldsymbol{\mu}_t + \frac{1}{2}W^\top\Sigma_t W}\left[(\boldsymbol{\mu}_t + \Sigma_t W)(\boldsymbol{\mu}_t + \Sigma_t W)^\top + \Sigma_t\right] \tag{30}$$

$$\frac{\partial^2\mathbb{E}_q\left[\log(p(\mathbf{x}, \mathbf{y}, \mathbf{z}|\boldsymbol{\theta})\right]}{\partial h\partial W} = -e^{h + W^\top\boldsymbol{\mu}_t + \frac{1}{2}W^\top\Sigma_t W}\left(\boldsymbol{\mu}_t + \Sigma_t W\right) \tag{31}$$

$$\frac{\partial^2\mathbb{E}_q\left[\log(p(\mathbf{x}, \mathbf{y}, \mathbf{z}|\boldsymbol{\theta})\right]}{\partial h^2} = -e^{h + W^\top\boldsymbol{\mu}_t + \frac{1}{2}W^\top\Sigma_t W} \tag{32}$$

where $l = 1, \ldots, M$ and $t = 1, \ldots, T$ are the trial and time indexes respectively.

---

[7] $\boldsymbol{\theta} = \{\mathbf{W}^{(0/j,j)}, \mathbf{h}^{(j)}, \mathbf{C}, \mathbf{d}, \boldsymbol{\Psi}, \tau^{(j)}\}$

## A.5 Learning the GP time constants

GP hyperparameters (time constant) are learned by gradient based numerical optimization of the joint log-likelihood. Following the notation of the main text we set, $\lambda_i^{(j)} = -\log(2\tau_i^{(j)})$, and we define a kernel $\mathbf{K}_i^{(j)} : \mathbb{R} \longrightarrow \mathbb{R}^{T \times T}$ such that, $\left[\mathbf{K}_i^{(j)}(\lambda)\right]_{ts} = \exp\left(-e^\lambda (t-s)^2\right)$.

The objective function takes the form,

$$\mathbb{E}_q\left[\log(p(\mathbf{x}, \mathbf{y}, \mathbf{z}|\boldsymbol{\theta})\right] = \sum_{l,j,i} -\text{trace}\left(\boldsymbol{K}_i^{(j)-1}(\lambda_i^{(j)})\mathbb{E}_q[\boldsymbol{z}_i^{(j)}\boldsymbol{z}_i^{(j)\top}]\right) - \log|\boldsymbol{K}_i^{(j)}(\lambda_i^{(j)})| + \text{const},$$

where $j = 0, ..., n$ is the latent factor, $l = 1, ..., M$ is the trial number and $i = 1, ..., d_j$ is the component of $\boldsymbol{z}^{(j)}$. Using the chain rule we obtain,

$$\frac{\partial \mathbb{E}_q\left[\log(p(\mathbf{x}, \mathbf{y}, \mathbf{z}|\boldsymbol{\theta})\right]}{\partial \lambda_i^{(j)}} = \text{trace}\left(\frac{\partial \mathbb{E}_q\left[\log(p(\mathbf{x}, \mathbf{y}, \mathbf{z}|\boldsymbol{\theta})\right]}{\partial \boldsymbol{K}_i^{(j)}}^\top \cdot \frac{\partial \boldsymbol{K}_i^{(j)}}{\partial \lambda_i^{(j)}}\right),$$

with

$$\frac{\partial \mathbb{E}_q\left[\log(p(\mathbf{x}, \mathbf{y}, \mathbf{z}|\boldsymbol{\theta})\right]}{\partial \boldsymbol{K}_i^{(j)}} = \frac{1}{2}\sum_l \left(-K_i^{(j)-1} + K_i^{(j)-1}\mathbb{E}_q[\boldsymbol{z}_i^{(j)}\boldsymbol{z}_i^{(j)\top}]K_i^{(j)-1}\right)$$

$$\frac{\partial\left[K_i^{(j)}\right]_{ts}}{\partial \lambda} = -e^\lambda (t-h)^2 \exp\left(-e^\lambda(t-s)^2\right).$$

## A.6 Parameter initialization

**Factorized TAME**. Before running EM on the full TAME, we obtain initial condition for the model parameters (all except the GP kernel hyperparameters) by means of running five iterations of EM for the temporally factorized version of the model. In particular, we replace the GP-prior over the latents with a product of a Gaussian normal distributions, i.e. $p(\boldsymbol{z}_i^{(j)}) = \prod_t p(z_{it}^{(j)})$, and $p(z_{it}^{(j)}) \sim \mathcal{N}(0, 1)$.

Under this prior assumption the joint likelihood as a whole factorizes over the temporal axis (i.e. the observations are temporally independent given the latents). As a consequence, the Hessian matrix of the joint pdf is sparse, and can be stored and inverted efficiently, allowing for the implementation of a full Newton scheme to numerically optimize for the MAP estimate of the posterior ever latents $\boldsymbol{z}$.

The EM-based optimization of the factorized TAME also needs an initial choice for parameters. We found empirically that a CCA-based heuristic works well for this purpose. Specifically, we set:

- $W^{(0,j)}$ to the first $d_0$ canonical directions $V$ between the square-rooted, mean-centered spike counts of population $j$, $\boldsymbol{s}^{(j)} = \sqrt{\boldsymbol{x}^{(j)}} - \mu_j$ and the task variables $\boldsymbol{y}$ ($\mu_j$ is the empirical mean of the square-rooted spikes).
- $W^{(j,j)}$ as the first $d_j$ principal direction for the orthogonal complement of the counts w.r.t the canonical directions, $\boldsymbol{s}_{\text{ort } t}^{(j)} = \boldsymbol{s}_t^{(j)} - V^\top V \boldsymbol{s}_t^{(j)}$. This will initially enforce orthogonality in the task relevant and private latent subspaces.
- $\boldsymbol{h}^{(j)}$ was set to the log of the empirical mean of the counts.
- $C$ was set to the first $d_0$ canonical directions $U$ between $\boldsymbol{s}$ and the square-rooted counts from all the neural populations, $\boldsymbol{Y} = [\boldsymbol{y}^{(1)}; \ldots; \boldsymbol{y}^{(m)}]$.
- $\boldsymbol{d}$ was set to the empirical mean of $\boldsymbol{s}$, and $\boldsymbol{\Psi}$ to the empirical covariance.

**GP time constants.** The initial GP time constants were drawn from a uniform random distribution $\tau_i^{(j)} \sim \text{U}[0, 0.5]$.

## A.7 Comparison of TAME-GP and SNP-GPFA

We compared our framework to that of SNP-GPFA Keeley et al. (2020), which identifies shared fluctuation between two neural populations, under the assumption of trial repeats with a common

stimulus-driven mean (corresponding to a dimensionality reduced peristimulus time histogram, or PSTH).

Briefly, the multi-area SNP-GPFA assumes that the spike counts of two areas, area A and area B, are generated according to

$$
\begin{bmatrix} \boldsymbol{Y}_j^A \\ \boldsymbol{Y}_j^B \end{bmatrix} = \text{Poisson} \left( f \left( \boldsymbol{W}_s \boldsymbol{X}^s + \begin{bmatrix} \boldsymbol{W}_{AA} & \boldsymbol{0} \\ \boldsymbol{0} & \boldsymbol{W}_{BB} \end{bmatrix} \begin{bmatrix} \boldsymbol{X}_j^{A,n} \\ \boldsymbol{X}_j^{B,n} \end{bmatrix} \right) \right), \tag{33}
$$

with $\boldsymbol{Y}_j^{A/B}$ the spike counts of population $A$ and $B$ for trial $j$, $f$ the soft-max non-linearity; $\boldsymbol{X}_j^{A/B,n}$ are drawn from a GP with factorized RBF covariance, which captures within area co-fluctuations for trial $j$; $\boldsymbol{X}^s$ corresponds to draws from another GP, which is shared across trials and populations, thus capturing the shared across area co-fluctuations.

We generated spike counts from the graphical model in figure S4A assuming a fixed trial duration (necessary for the SNP-GPFA), in different conditions: 1) fixing the shared dynamics across trials (as in SNP-GPFA, figure S4B, top), or 2) varying the shared dynamics across trial (figure S4B, bottom).

Specifically, for the first condition the counts followed (33), but replacing the non-linearity with an exponential. For the second case, the counts follow Poisson statistics of the form

$$
\begin{bmatrix} \boldsymbol{Y}_j^A \\ \boldsymbol{Y}_j^B \end{bmatrix} = \text{Poisson} \left( \exp \left( \boldsymbol{W}_s \boldsymbol{X}_j^s + \begin{bmatrix} \boldsymbol{W}_{AA} & \boldsymbol{0} \\ \boldsymbol{0} & \boldsymbol{W}_{BB} \end{bmatrix} \begin{bmatrix} \boldsymbol{X}_j^{A,n} \\ \boldsymbol{X}_j^{B,n} \end{bmatrix} \right) \right), \tag{34}
$$

where we added a trial dependency to the shared Gaussian process factor.

We set the dimensionality of the shared factor to 2, and of each private factors to 3. We simulated spike counts from two populations of 30 neurons for 50 trials, each having 100 time points with a 0.05 second resolution. The average firing rate of both population was set to 10Hz. We fit the simulated spike counts with TAME-GP and SNP-GPFA for both conditions. The results show that TAME-GP captures the between area co-fluctuation in both scenarios while SNP-GPFA fails when the shared dynamics vary between trials, as expected by the model assumptions (Fig. S4C,E). We assessed the accuracy of the factorization of the spike-count variance by means of Lasso regression. In particular, we regressed the ground truth latents from the estimated latents of the different models, and quantified regression goodness-of-fit in terms of cross-validated $R^2$ (Fig. S4D,F). We quantified the contribution of each latent factor to the regression in terms of the magnitude the associated coefficients. Results (reported in Table S1) show that 1) both models can factorize the variance when the shared dynamics are fixed across trials, with SNP-GPFA achieving a cleaner decomposition (expected given that it is a closer model of the true data generating process in this case); 2) TAME-GP achieves a near optimal factorization when the shared latents vary across trials (as assumed by its generative model), while SNP-GPFA is unable to find the appropriate decomposition. Overall, TAME-GP estimator proves more robust to deviations from its underlying model assumptions.

### A.8 Selecting the number of private and shared dimensions in real data

We select the number of private and shared dimensions to fit in real data by optimizing these hyperparameters via a grid search. A priori we set the maximum number of dimensions to be evaluated as the number of PCs needed to account for 80% of the population variance (in this case, 5 dimensions). Fig.S6 shows estimates of model fit quality as a function of the number of dimensions included in private and shared latents for the multi-area TAME-GP presented in Fig.4I-K. The results show a well-behaved cross-validated $R^2$ landscape, with optimal dimensionalities $(5, 5)$.

### A.9 Fitting TAME-GP to monkey premotor and motor responses during reaches

We also tested our estimator on a publicly available dataset Perich et al. (2018) that records neural activity in premotor cortex (PMd) and primary motor cortex (M1) of macaques during sequential

reaches (binned at 10ms resolution). Specifically, the monkey controls an on-screen cursor and is rewarded for moving that cursor to an indicated reach target, with multiple targets presented in a trial. Since there are minimal kinematic requirements for the reaching movements (e.g., very brief hold times), the monkey typically makes relatively smooth series of reaches. As pPCA latent structure was very poor quality in this dataset, we restricted our comparison between TAME-GP, with task manifold aligned to screen position, and PGPFA, with latent dimensionality d=2 (Fig.S7). Visually, the latent structure extracted by TAME-GP seems to better capture the animal behavior, so we asked (in $R^2$ terms)[8] how much information about the task variables can be linearly decoded from their respective latents for TAME-GP and PGPFA with variable latent dimensionality (Fig. S7C, F). These results confirms that in this dataset as well, the TAME-GP task aligned manifold provides a compact account of neural variability with respect of task variables of interest, which does not align with the overall axes of neural variability of the data. PGPFA needs substantially higher dimensional latent spaces (10d vs. 2d) to capture the same amount of task-relevant neural variability.

## A.10    SUPPLEMENTARY FIGURES

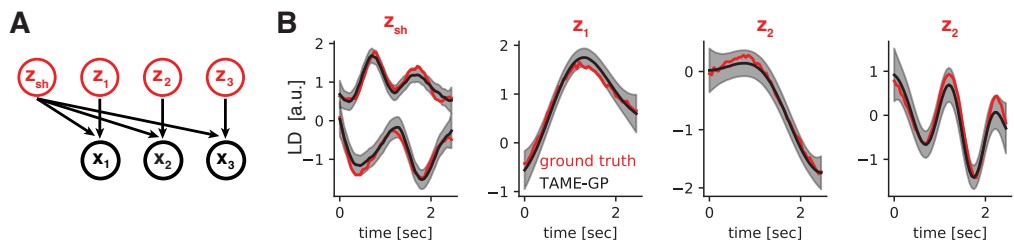

Figure S1: Multi-area parameter reconstruction. **A**. TAME-GP generative model for three brain areas with shared interactions. **B**. **D**- Latent variables estimation for within model simulated data: ground truth latent factors and model posterior mean $\pm$ 95% CI for all latent dimensions.

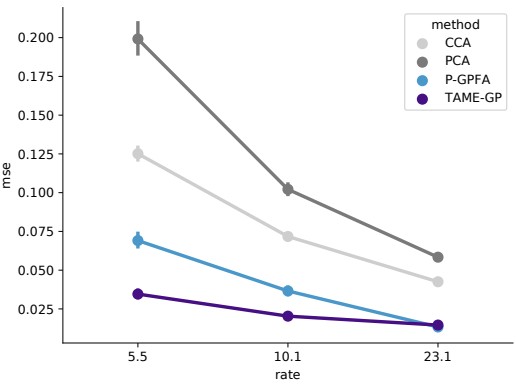

Figure S2: Task-aligned latent dynamics reconstruction (extends Fig. 2C). Mean squared error between the true task relevant dynamics and the model reconstruction based on the 2 dimensional task relevant latent factor for CCA and TAME-GP, and the full 6 dimensional latent space for P-GPFA and PCA. In contrast, figure 2C shows the MSE based only on the first 2 principal latents for pCCA and P-GPFA; Error bars represent the mean $\pm$ s.d. over 10-fold cross-validation.

---

[8]All decoding used Lasso regression, with 5-fold cross-folding for hyperparameter estimation.

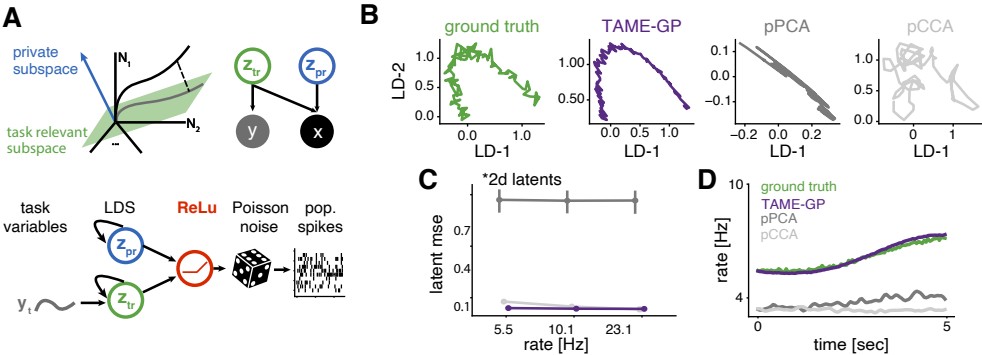

Figure S3: Effects of model mismatch on latent structure estimation. **A**. TAME-GP graphical model for single area (top) and schematic for data generating process (bottom). LDS dynamics where transformed into firing rates through a ReLu non linearity (replacing the exponential used in Fig. 2), so that the latents have now an additive instead of multipliative effects on the observed neural activity. **B**. Ground truth task relevant dynamics (green) and estimated low dimensional projection for TAME-GP (purple), pPCA (dark gray) and pCCA (light gray).**C** Mean squared error between the true shared dynamics and the model reconstruction, mean $\pm$ s.d. over 10-fold cross-folding. **D**. Example single trial firing rate reconstruction.

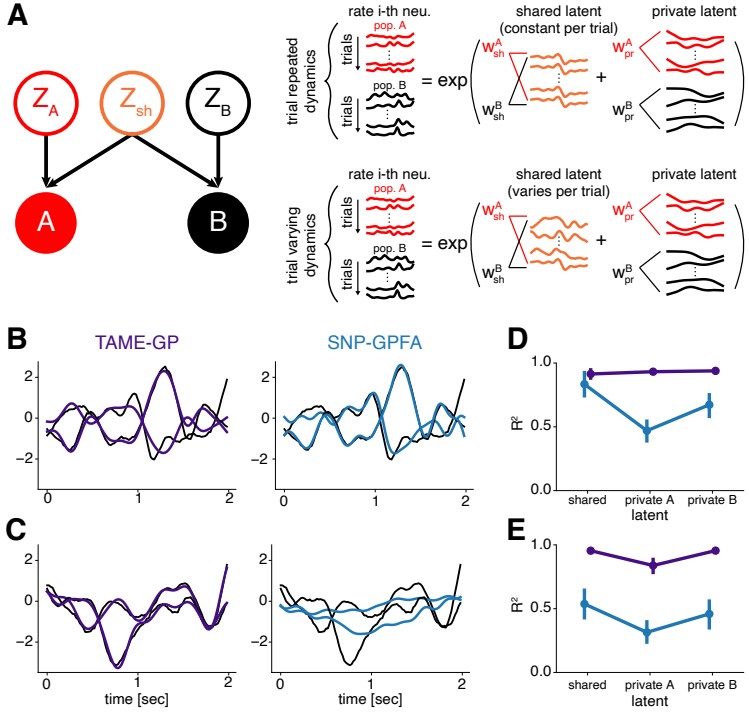

Figure S4: Communication subspace estimation with SNP-GPFA and TAME-GP (extends Fig. 3). **A**. Scheme of the spike count generative model for trial repeated (top right) and trial varying (bottom right) shared dynamics. **B,C**. Ground truth shared dynamics (black lines) and model reconstructions (colored lines) for the trial repeated (**B**) and trial varying (**C**) conditions. **D,E**. Ground truth shared and private dynamics variance explained by model predictions for the trial repeated (**D**) and trial varying (**E**) conditions.; error bars represent mean $\pm$ standard deviation over a 5-fold cross validation.

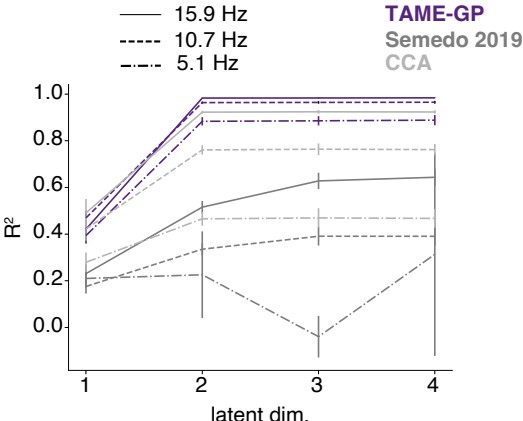

Figure S5: Model fit of shared and task aligned dynamics. R2 of the linear regression between the ground truth task aligned latent dynamics and the model MAP estimate for TAME (purple), PCCA (light grey) and reduced rank regression (dark grey). Extends fig. 3I in the main text to multiple average firing rates.

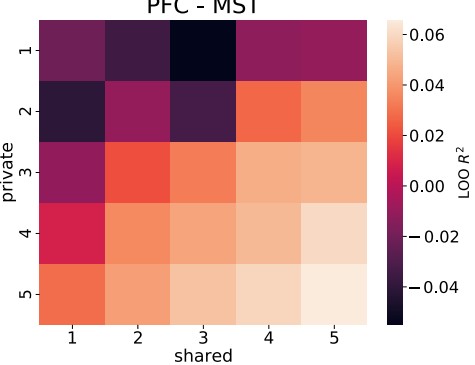

Figure S6: Latent dimensionality selection, for the MSTd and dlPFC communication manifold analysis (extends fig. 4I-K). Heat-map of the leave-one-neuron-out R2 of the spike count variance explained by a TAME-GP for different combination of shared and private latent dimensions. The upper bound on dimensionality was set to the number of principal components needed to explain 80% of the population spike count variance.

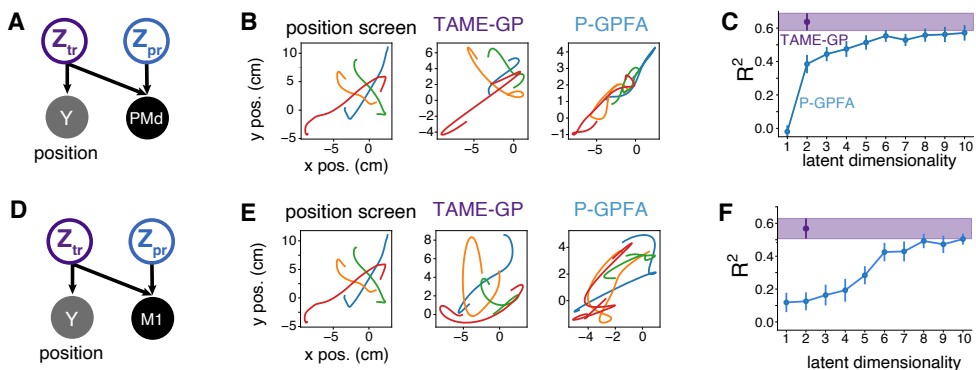

Figure S7: TAME-GP manifold estimation for monkey premotor (PMd) and motor (M1) neural responses during reaching. **A**. Graphical model for PMd neural manifold aligned to 2d coordinates of hand on screen. **B**. Behavior and corresponding PMd latent population trajectories for four example individual reaches (colors), extracted with TAME-GP and P-GPFA. **C**. Lasso regression decoding of position; TAME-GP $R^2$ (purple) is based on a 2d task relevant latent. P-GPFA $R^2$ (blue) estimates were obtained for a range of latent dimensions (1-10). **D, E, F**. Same as A, B, C for M1.

## A.11 SUPPLEMENTARY TABLE

| Lasso results | | | | |
|---|---|---|---|---|
| **model** | **sim type** | **ground truth latent** | **model latent** | $\|\beta\|$ |
| SNP-GPFA | fixed | private A | private A | 0.252458 |
| | | | private B | 0.008377 |
| | | | shared | 0.065025 |
| | | private B | private A | 0.001043 |
| | | | private B | 0.373415 |
| | | | shared | 0.012722 |
| | | shared | private A | 0.014438 |
| | | | private B | 0.026413 |
| | | | shared | 0.665547 |
| | variable | private A | private A | 0.128838 |
| | | | private B | 0.256987 |
| | | | shared | 0.044492 |
| | | private B | private A | 0.046482 |
| | | | private B | 0.332019 |
| | | | shared | 0.114382 |
| | | shared | private A | 0.058657 |
| | | | private B | 0.386729 |
| | | | shared | 0.010308 |
| TAME-GP | fixed | private A | private A | 0.222231 |
| | | | private B | 0.00472 |
| | | | shared | 0.16166 |
| | | private B | private A | 0.016308 |
| | | | private B | 0.419728 |
| | | | shared | 0.02365 |
| | | shared | private A | 0.1016 |
| | | | private B | 0.006841 |
| | | | shared | 0.476519 |
| | variable | private A | private A | 0.268177 |
| | | | private B | 0.011153 |
| | | | shared | 0.032323 |
| | | private B | private A | 0.003969 |
| | | | private B | 0.411102 |
| | | | shared | 0.020695 |
| | | shared | private A | 0.019493 |
| | | | private B | 0.005826 |
| | | | shared | 0.658865 |

Table S1: Lasso regression coefficients, related to session A.7. Norm of the coefficients of the Lasso regression between the ground truth latent dynamics and the SNP-GPFA/ TAME-GP predicted latents. Lasso hyperparameters are set by grid search with a 5-fold cross-folding procedure.

