# OpenReview forum: "A probabilistic framework for task-aligned intra- and inter-area neural manifold estimation"
_ICLR.cc/2023/Conference — ICLR 2023 notable top 25%_

### Official Review · Reviewer_7WyW · 2022-10-23

**Confidence:** 4
**Correctness:** 4
**Technical Novelty And Significance:** 4
**Empirical Novelty And Significance:** 4
**Recommendation:** 8

**Clarity, Quality, Novelty And Reproducibility:**

+ Clarity is one of the weaknesses of the draft. I thought that I departed with good background to fully understand the paper, both in terms of neuroimaging as well as in terms of maths, but I found myself struggling a bit to follow what was going on during my first read. Yes, one manages to decode the paper after several reads, but I wonder whether the ideas are a bit obfuscated if even someone with related skills needs several reads. Unfortunately, I do not have a better suggestion… I can see the issue but not the solution.
+ Quality: Good. There is adequate mathematical rigor and there are no major flaws that I can think of. The verification and validation efforts are sufficient in my opinion.
+ Novelty: The extension that TAME-GP represents over its predecessor pCCA is certainly not obvious or trivial.
+ Reproducibility: I believe that with a little bit of effort that I can certainly replicate the methods. The part of the experimental results are a bit less clear.


**Details Of Ethics Concerns:**

Two experiments on animal models are discussed (mice and monkeys). However, no ethical approval details are indicated.

**Strength And Weaknesses:**

Strengths
+ The novel way to generalize pCCA to non-Gaussian noise model.
+ Results do accompany; the new model not only captures more observed variance but also appears to better decode task-related information.
Weaknesses
+ It is unclear how the mathematical decisions match physiological and or functional rationales, which means that even if the method works, its interpretation remains cumbersome.
+ Conciseness….or the lack of it.


**Summary Of The Paper:**

A method (TAME-GP) to capture task-related variance in neural firings observations is presented. The method has its roots on probabilistic canonical correlation analysis (pCCA), but proposes an extension for non-Gaussian noise. This is achieved by replacing pCCA’s prior that assumes a normal distribution with a Gaussian process as well as adding an explicit hidden (latent) layer z for relating inputs x (spike counts) and the output y (task variables).
Two verification efforts on synthetic data and two validation efforts on experimental data are provided. Moreover, of the latter two one is of segregational nature (local activities), and the other of integrational nature (communication between regions).


**Summary Of The Review:**

+ Including the latent layer makes me wonder, how would hidden graphical models e.g. hidden Markov models or others sophisticated variants, perform compared to TAME-GP?
+ Regarding the aforementioned weakness on the relation of the mathematical decisions having a physiological or functional counterpart, why n+1 latent variables? Why Gaussian processes? Why in the expectation step not using a transformation  e.g. Anscombe or Freeman-Tukey or other, rather than approximating the posterior again with a normal distribution? What would be the physiological interpretation of the model hyperparameters (for instance, but not limited to; the linear approximation Cz^j +d, the \tau, etc)?
+ Has Procrustes also been applied to the other methods being compared in Fig 2B? It seems that at least P-GPFA and pPCA are not correctly aligned?
+ The order of presentation of some mathematical elements is confusing. For instance. Parameter K is introduced in pg 3 during the expectation step but not defined until the end of the maximization step pg 4.
+ As one of the advantages of TAME-GP over pCCA, it is stated that pCCA only considers pairwise relations. This somewhat implies that TAME-GP handles relations of arity higher than 2 between regions. The k_j terms are clearly pairwise among latent z variables. It is not clear to me how this propagates to brain circuits of more than two regions.

Minor details
+ Cross-validation: Popular terminology but mathematically inaccurate. Suggestion: internal validity using cross-folding.
+ In the discussion: Causality of the outcomes is claimed, but I can’t see how any of the traditional causal assumptions for any popular probabilistic causal definition (Spirtes-Glymour, Pearl, Suppe, etc) are shown or demonstrated here.
+ Appendices A.1.1 and A.1.2 are possibly unneeded. References should suffice.

---

> ### Author Response · Authors · 2022-11-18
> **Response to reviewer 7WyW**
>
> We thank the reviewer for the feedback. Below we include an itemized reply to comments, in the order in which they appear in the review:
>
> 1. We have included additional justification for the biological and statistical relevance of the model choices. Briefly, Poisson noise is the most sensible option, especially when looking at single trials within tens of ms temporal resolution; the exponential link function is justified by the history of statistical modeling in this field going back to GMLs, but also biologically -- by divisive normalization and documented multiplicative stochastic gain fluctuations in cortical data.
>
> 2. About options not considered in the current model comparison: Discrete latent space models as HMMs are typically not good models of cortical activity, although they can capture phenomena such as up-down states (e.g. Engel work) or underlying fixed point attractor dynamics structure (e.g. Tkacik work in the retina). Moreover, none of the previous approaches include task-relevant variables as part of the model. As such, we chose not to include them in a direct model comparison, sticking to more obvious alternatives, such as P-GPFA.
>
> 3. Regarding various model/estimator choices: why  n+1 latents: we are generally looking for partitioning of variability into shared and private components, but it is a valid question what should a shared manifold across more than 2 areas supposed to mean physiologically; one immediate application is to detect broadcasted signals (e.g. coming from subcortical or thalamic nuclei) that affect multiple cortical regions simultaneously.  Of course, the causal flow of information cannot be resolved based on correlation structure alone, but it may meaningfully restrict the hypothesis space. Why GPs: it is a mathematically convenient way of modeling dependencies over time; the use of neural dynamics specific kernels allows for additional interpretability at limited extra cost (work in progress). Why not a transform: low cortical rates, with <1 spikes/bin, means that the Gaussian assumption, even after nonlinearly transforming the data is not matched well.
> Parameter interpretation: Cz_j are the (shared) fluctuations around the mean d. \tau gives the decorrelation time constant of the process. Some of the choices were motivated by GPFA and P-GPFA, which are known to work well for modeling low-dimensional latent neural dynamics.
>
> 4. Procrustes alignments are used for all models, using data across all trial types; the main 2 axes of variability extracted by pPCA and P-GPFA mix both task variable aligned and task variable independent, and this impacts strongly the alignment at the individual trial level.
>
> 5. We've edited the text to alleviate some of the order confusion, including labeling the GP equations and back-referring to those later in the text.
>
> 6. The model allows for the same set of latent variables to cause covariability in more than two sets of neural observations;  it is ``not just pairwise'' in this sense.
>
> 7. We've adjusted the cross-validation references as requested. The term is now only used in the strict sense, for the leave-one-neuron-out procedure.
>
> 8. Causality: We were simply referring to the assumed causal structure implied by the graphical model. We have revised the text to remove any particular terminological associations with causality in the formal sense.
>
> 9. While we agree that the appendices A1.1,2 are not strictly needed, we feel that outlining the basics of  probabilistic PCA and CCA may be still useful for the neuroscience readers, so we decided to keep them.
>
> 10. Animal experiments details: We did not report the detailed ethics regulations followed for the experiments as it is hard to do without breaking the double blind review. This information will be included in the published version.

---

### Official Review · Reviewer_1fJi · 2022-10-24

**Confidence:** 4
**Correctness:** 3
**Technical Novelty And Significance:** 3
**Empirical Novelty And Significance:** 2
**Recommendation:** 6

**Clarity, Quality, Novelty And Reproducibility:**

This paper has decent clarify and quality.
The novelty is a bit incremental.

**Strength And Weaknesses:**

****Pros:
- The paper is generally well-written and, for the most part, clear.
- Understanding inter-area communications represents an interesting problem in neuroscience. This is generally a less studied topic, but has attracted more attention recently due to the increased experimental capacity to monitor the activities from multiple regions.
- The proposed method is pretty straightforward.
- Several applications were considered, based on both simulations and neuroscience data.


****Cons:
- The way proposed in the paper to model the communication is not totally convincing. The signals across the different regions are only allowed to interact via a multiplicative modulation. It is unclear whether this is reasonable for neural systems.
- The technical contribution presented in this paper is rather incremental. The model is largely linear and assembles multiple components from prior work. Most of the components are fairly standard. The main innovation seems to the interpretation of Eq (1) as a multi-region communication model (but see my next comment).
- It is unclear what’s special about the inter-area communication. Eq. (1) could also be interpreted as a model for a single brain area with increased latent dimension. This needs to be further clarified.
- The real neuroscience data applications are detailed but unconvincing. Maybe the authors could articulate better what new insights were revealed. This is particularly important given the rather incremental technical contribution.
- The authors showed examples in which case the model could be recovered. Are there also cases for which the model components can not be recovered? This seems to be an important question for interpreting the output of the model, but it is studied in the paper.


Other comments:
- Practically, how to set the number of latent dimensions in each area?
- The title should be changed. It is unclear what “INTRA- AND INTER-AREA NEURAL MANIFOLD” really means. I understand that “neural manifold” is good for branding, but that should come with the cost of sacrificing the accuracy and faithfulness of the title.
- It is unclear how to think about the mouse ADN recording in the context of inter-area or intra-area communication. This needs a clarification.
- There should be a more thorough treatment of the literature on using the task variable to guide the estimation of latent variables. A few papers that would be useful to discuss:
Sani, Omid G., et al. "Modeling behaviorally relevant neural dynamics enabled by preferential subspace identification." Nature Neuroscience 24.1 (2021): 140-149.

Zhou, D., et al "Learning identifiable and interpretable latent models of high-dimensional neural activity using pi-VAE." Advances in Neural Information Processing Systems 33 (2020): 7234-7247.

Hurwitz, Cole, et al. "Building population models for large-scale neural recordings: Opportunities and pitfalls." Current Opinion in Neurobiology 70 (2021): 64-73.

- Fig 1a. The model variables need to be better explained in the figure caption/legend.



**Summary Of The Paper:**

In this submission, the authors proposed a method to partition neural response variability within and across brain areas.

**Summary Of The Review:**

On the one hand, it’s a decent paper. I don’t see fatal errors in this paper. However, I also don’t feel the work to be too exciting, so perhaps not quite up to par for ICLR.  Overall, I consider this paper to be slightly below the border of acceptance. But I’d also be ok if this paper gets accepted.

---

> ### Author Response · Authors · 2022-11-18
> **Response to reviewer 1fJi**
>
> Thank you for the feedback. Below we reply to the various comments and concerns in the order in which they appear in the review.
>
> 1) Multiplicative modulation in the model is a standard assumption in many latent state models commonly used in the neuroscience literature (P-GPFA, Poisson LDS, etc). From a statistical perspective, it is a natural parameterization for a Poisson observation model; from a neuroscientific perspective, multiplicative modulation of neural responses is a ubiquitous phenomenon, believed to play key roles in a variety of cortical computations (see e.g. review by Carandini and Heeger 2012, or the recently documented functional role of multiplicative modulation in task-specific information routing,  Haimerl et al, 2021).
> To assess the importance of the multiplicative effects assumption, we've included a new set of simulation results where the ground truth data structure involves additive rather than multiplicative latent interactions and evaluated the effects of such model mismatch on the TAME-GP extracted structure. We found that, although the mismatch does affect the quality of the estimates in quantitative terms, the nature of the extracted structure still reflects the structure of the data in qualitative terms, so it is still yielding interpretable latents. This is true for a range of hyperparameter values. The model could (and should) break if the task aligned subspace is perfectly aligned to the private manifold; however in high dimensional spaces (like the neural space) it is highly unlikely, subspaces tend to be orthogonal. Moreover, it was shown (Semedo, 2009) that, at least for V1 and V4, the communication subspace is orthogonal to the private subspaces of the areas.
>
> 2-3. The multi-area interaction model specifically assumes that some of the latents are private whereas others are not (i.e. part of the communication manifold). We would argue that putting familiar elements together in a new way is the essence of much of statistical probabilistic modeling and it has yielded several success stories in neuroscience (e.g. one could argue that GPFA is just GP regression, but that does not diminish the success of GPFA and its later variants in explaining neural data across species and experimental setups).
>
> 4) Given the ICLR format, we see the role of this manuscript as primarily describing in technical terms the statistical approach, while the core scientific insights derived from using it on unpublished data were reserved for a separate journal publication. What the neural analyses show at this stage is primarily the fact that this statistical model is a more precise description of the population responses than traditional alternatives. Nonetheless, we also find a previously unknown flow of task-relevant information between areas MST and dlPFC in the context of the firefly task, which is surprising given that the two areas do not share direct anatomical connectivity. This points to the beliefs about distance to target emerging as a distributed computation across multiple areas.
>
> 5) In our simulations, the TAME-GP estimator proved robust to a range of deviations from model assumptions (see new Suppl.Fig. using RELU nonlinearities and additive interarea interactions), for a range of parameter setups. While there are some extreme situations in which the estimator fails (perfectly aligned axes for noise and signal), we would not consider those to be likely or biologically relevant. Finally, while the additive dataset reduces the model mismatch of the linear models (dPCA,CCA), the ability of TAME to match the Poisson noise and temporal regularities in the data allows it to still win the direct model comparison, albeit by a smaller margin.
>
> 6) The number of latents is generally determined by model comparison.
>
> 7) We feel that the title does actually reflect the primary goal of TAME-GP of extracting within area and across-area latent structure, so we decided not to change the title.
>
> 8) We use the mouse ADN as a first test of latent structure reconstruction in neural data where the underlying code has a known ring structure; you can think of it as a data with biological ground truth latents.
>
>
> 9) We thank the reviewer for the extra references, though we find  some to be tangential to the core topic. We now cite Hurwits2021 in the intro. The Sani model is essentially a Kalman filter with a sequential estimation procedure. Like dPCA, the Zhou reference does not consider dependencies over time -- a core aspect of neural population responses-- but it does allow for  complex nonlinear embeddings of population activity vectors; now mentioned in the related work.
>
> 10) We've worked on improving the legend captions.

---

> > ### Comment · Reviewer_1fJi · 2022-11-28
> > **thanks for the response**
> >
> > I would like to thank the authors for their response to my comments. The response adddressed some of my concerns. In particular, the addition simulation for point 1 was useful. I have increased my score from 5 to 6.

---

### Official Review · Reviewer_4RNV · 2022-10-25

**Confidence:** 4
**Correctness:** 4
**Technical Novelty And Significance:** 3
**Empirical Novelty And Significance:** 2
**Recommendation:** 8

**Clarity, Quality, Novelty And Reproducibility:**

The paper is well-written and the figures are laid out nicely. The performance of the current method is good compared to some popular methods from the same class. One of the real datasets in the current study is publicly available which will benefit reproducing the work. Will authors release their code if accepted?  Latent models from similar classes are available. But the current study is novel as it combines several dimensionality reduction approaches and achieved better performance and interpretability.

**Strength And Weaknesses:**

Strength:
1.	The authors have a good sense in demonstrate the interpretability and application of the new model using various simulation data. I really like the part about the communication subspace (figure 3). The fact that the current model achieved better performance compared to Semedo 2019 and Keeley 2020 for both simulation datasets with or without trial repeats is exciting.
2.	The authors have compared the current model with multiple popular models in fitting both simulation and real datasets, which makes the overall results more convincing.

Weakness:
1.	Some recent RNN-based latent models eg. LFADs and Oerich 2020, were overlooked in the current manuscript. It would be great to discuss those.
2.	It is not clear to me whether such a model could generate novel knowledge or testable hypothesis about neuron data.

**Summary Of The Paper:**

Neuroscientists are generating rich datasets with population recordings from multiple brain areas simultaneously while the animal is engaged in a complicated behavior. It urges for an analytical tool to extract interpretable information from these rich datasets. Latent variable models are popular approaches for generating a compressed summary of such data. However, the interpretability of the latent model is limited. In the current paper, the authors presented a novel latent variable model which combines dPCA and pCCA into a graphic model to learn task or stimulus-relevant latent variables of neuron population data. The author demonstrated the model successfully learns interpretable latent variables that could account for both inter- and intra- area variance. Overall, the current model achieved superior performance compared to several population models. The message of the paper is clear and the results are solid to me.

**Summary Of The Review:**

Overall, I recommend accept the current work as the authors have not only developed a new model, but also carried out thorough study on the model.

---

> ### Author Response · Authors · 2022-11-18
> **Reply to reviewer 4RNV**
>
> Thank you for the feedback.
>
> 1. We consider LFADS only tangentially related to this work since they do not directly provide dimensionality reduction, but we've added a mention to these methods in the discussion.
>
> 2. About the interpretability of the extracted structure: extracting causal structure is arguably always a challenge of purely statistical approaches; TAME-GP can detect previously unknown structure about the across area variability which relates to meaningful task features. When paired with known anatomical structure, this can in turn be used to generate new scientific hypotheses. Concretely, in the context of our own results the method revealed a previously unknown flow of task-relevant information between areas MST and dlPFC, which do not share direct anatomical connectivity. This points to the beliefs about distance to target emerging as a distributed computation across multiple areas.
>
> 3. The code is on github and will be linked from the manuscript on publication.

---

### Official Review · Reviewer_RPBA · 2022-10-25

**Confidence:** 2
**Correctness:** 4
**Technical Novelty And Significance:** 4
**Empirical Novelty And Significance:** 3
**Recommendation:** 8

**Clarity, Quality, Novelty And Reproducibility:**

A few comments to improve the clarity of presentation:
In Eq.  (1), what is the subscript i?
In the third line below Eq. (1), "to introduces" should be "to introduce".
In Section 3, first line, "due the Poisson noise" should be "due to the Poisson noise".

**Strength And Weaknesses:**

Strengths:
The paper is clealy presented with excellent use of English. The technical contributions are significant. The pros and cons as well as the broader impact of the proposed method are discussed in depth.

Weaknesses:
In Section 2, the notations may need to be explained a bit more.


**Summary Of The Paper:**

This paper proposes a probabilistic approach for learning interpretable task-relevant neural manifolds that capture both intra- and inter-area neural variability with single trial resolution. Task Aligned Manifold Estimation with Gaussian Process priors (TAME-GP) incorporates elements of demixed PCA and probabilistic CCA into a graphical model that additionally includes biologically relevant Poisson noise. The model uses a Gaussian Process (GP) prior to enforce temporal smoothness, which allows for robust reconstruction of single-trial latent dynamics. Experiments using both synthetic data and neural recordings from rodents and primates during naturalistic tasks demonstrate the robustness and flexibility of TAME-GP in comparison to alternative approaches.

**Summary Of The Review:**

This is a solid piece of work. The paper has solid technical contributions which have been adequately demonstrated through extensive experiments and indepth discussions. The proposed TAME-GP could be very useful as solution/tool in practice for dissecting sources of variability within and across brain areas during behavior.

---

> ### Author Response · Authors · 2022-11-18
> **Reply to reviewer RPBA**
>
> Thank you for the feedback. We’ve clarified the notation and corrected the
> typos in the updated manuscript.

---

### Decision · Program_Chairs · 2023-01-20

**Decision:**

Accept: notable-top-25%

**Justification For Why Not Higher Score:**

The real neuroscience data applications are detailed but unconvincing. Maybe the authors could articulate better what new insights were revealed. This is particularly important given the rather incremental technical contribution.

**Justification For Why Not Lower Score:**

All reviewers acknowledged that this work is solid with extensive experiments and in depth discussions.

**Metareview: Summary, Strengths And Weaknesses:**

Summary: This paper proposes a probabilistic approach for learning interpretable task-relevant neural manifolds that capture both intra- and inter-area neural variability with single trial resolution. Experiments using both synthetic data and neural recordings from rodents and primates during naturalistic tasks demonstrate the robustness and flexibility of TAME-GP in comparison to alternative approaches.

Strengths: The paper has solid technical contributions which have been adequately demonstrated through extensive experiments and indepth discussions. The proposed TAME-GP could be very useful as solution/tool in practice for dissecting sources of variability within and across brain areas during behavior. Understanding inter-area communications represents an interesting problem in neuroscience.

Weakness: The technical contribution presented in this paper is rather incremental. The model is largely linear and assembles multiple components from prior work. Most of the components are fairly standard.

**Note From Pc:**

if the above contains the word "oral" or "spotlight" please see: "oral" presentation means -> notable-top-5% and "spotlight" means -> notable-top-25%. As stated in our emails, we are disassociating presentation type from AC recommendations